# Revisiting Chain-of-Thought in Code Generation:
# Do Language Models Need to Learn Reasoning before Coding?

**Ren-Biao Liu** [1 2]  **Anqi Li** [1 2]  **Chaoding Yang** [1 2]  **Hui Sun** [1 2]  **Ming Li** [1 2]

## Abstract

Large Language Models (LLMs) have demonstrated exceptional performance in code generation, becoming increasingly vital for software engineering and development. Recently, Chain-of-Thought (CoT) has proven effective for complex tasks by prompting LLMs to reason step-by-step and provide a final answer. However, research on *how LLMs learn to reason with CoT data for code generation* remains limited. In this work, we revisit classic CoT training, which typically learns reasoning steps before the final answer. We synthesize a dataset to separate the CoT process from code solutions and then conduct extensive experiments to study how CoT works in code generation empirically. We observe counterintuitive phenomena, suggesting that the traditional training paradigm may not yield benefits for code generation. Instead, training LLMs to generate code first and then output the CoT to explain reasoning steps for code generation is more effective. Specifically, our results indicate that a 9.86% relative performance improvement can be achieved simply by changing the order between CoT and code. Our findings provide valuable insights into leveraging CoT to enhance the reasoning capabilities of CodeLLMs and improve code generation.

## 1. Introduction

Code generation (Jiang et al., 2024; Roziere et al., 2023; Hui et al., 2024) aims to automatically generate high-quality, executable programs that satisfy specific problem specifications. Recently, Large Language Models (LLMs), trained on vast amounts of natural language and code, have demon-

[1]National Key Laboratory for Novel Software Technology, Nanjing University, Nanjing, China [2]School of Artificial Intelligence, Nanjing University, Nanjing, China. Correspondence to: Ming Li <lim@lamda.nju.edu.cn>.

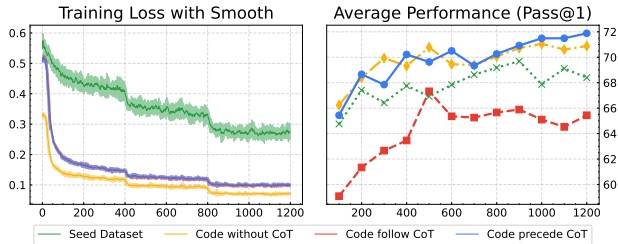

Figure 1: The smoothed training loss curve and the average performance across benchmarks of different SFT strategies.

strated significant potential in understanding and generating complex programs (Brown et al., 2020; Askell et al., 2021; Dubey et al., 2024; Guo et al., 2025), thereby attracting considerable attention. However, training large-scale LLMs with many learnable parameters is often highly time- and resource-intensive (Han et al., 2024). As a result, LLMs need to improve computational efficiency and task performance when adapting to specific downstream domains (Zhu et al., 2024), including code generation (Austin et al., 2021).

Supervised Fine-Tuning (SFT) (Bai et al., 2022; Stiennon et al., 2020) offers a promising solution to enhance the specific capabilities of pre-trained LLMs for downstream tasks. Prior studies (Zhou et al., 2023; Gunasekar et al., 2023) suggest that high-quality training datasets for SFT can lead to significant performance gains with fewer time and resource requirements. Specifically, Zelikman et al. (2022) and Hsieh et al. (2023) emphasize that detailed Chain-of-Thought (CoT) (Wei et al., 2022) can improve data quality, leading to enhanced performance in complex reasoning tasks, including math (Guan et al., 2025; Ho et al., 2023), logic (Xu et al., 2024; Huang & Chang, 2023), and commonsense (Li et al., 2024a; 2023b; 2022). CoT is an effective technique proposed to encourage LLMs to generate intermediate reasoning steps before outputting the final answer. SFT with CoT data enables the model to provide more reliable answers through thoughtful consideration and explanation, as empirically (Ye et al., 2024; Yu, 2024) and theoretically (Feng et al., 2023; Liu et al., 2024b) confirmed.

Code generation task also relies on the model's rigorous and complex reasoning ability (Petty et al., 2024; Shen & Zhang,

2024). However, the research about how LLMs learn to reason with CoT data for code generation in SFT remains limited. Therefore, this work revisits classic CoT training, which typically organizes reasoning steps before the final answer. We aim to investigate how CoT reasoning affects the code generation performance of models in the SFT stage and identify the underlying reasons for these effects.

Due to the absence of datasets in a unified format, we first construct a diverse, high-quality, and compact dataset that includes natural language descriptions of program questions, reasoning processes, and code solutions. This dataset enables an analysis of the behavior and capabilities of language models during the SFT phase, providing a robust foundation for our study. We conduct SFT experiments using different base models with the enriched dataset of 50K data pairs. This approach enables independent and objective conclusions unaffected by any specific pre-training dataset. Additionally, we examine key parameters during the SFT process to ensure consistency in our findings across different configurations. Our result in Figure 1 shows that we can significantly improve performance by changing the order between CoT and code. The findings suggest that high-quality code is already suitable to act as CoT. At the same time, traditional CoT should be regarded as an explanation of the code rather than a description of the reasoning process. We provide a comprehensive analysis to investigate the behavioral patterns of LLM in code generation and further examine the proposed findings. In summary, the main contributions of this work are as follows:

- **Synthetic Dataset** We build a high-quality and well-organized dataset of 50K pairs optimized for code generation through seed collecting and data synthesis.

- **Strategies Study** We demonstrate that high-quality code can serve as a practical CoT, while traditional CoT should be considered an explanation.

- **Key Insights** We uncover how CoT position and dataset pattern influence performance, providing essential insights and confirming our findings.

The remainder of this paper is organized as follows: Section 2 reviews recent studies relevant to our work. Section 3 presents the preliminaries required to understand the background. Section 4 provides detailed information about implementing our empirical study, including data collection and training strategies. Sections 5 and 6 present the main results and further discussion, respectively. Section 7 concludes the paper and outlines future work.

## 2. Related Work

**Large language models** Large Language Models (LLMs) (Brown et al., 2020; Achiam et al., 2023; Zhao et al., 2023)

have emerged as powerful tools in Natural Language Processing (NLP), advancing language understanding and generation. LLMs (Radford et al., 2019; Askell et al., 2021; Bai et al., 2022) exhibit robust generalization capabilities due to large-scale pre-training. However, due to the distribution gap between code data and the general text corpora (Huang et al., 2024), their code generation performance remains limited, leading to difficulties in logical reasoning and syntactic precision (Jimenez et al., 2024; Dou et al., 2024). Consequently, CodeLLMs (Roziere et al., 2023; Bai et al., 2023) are trained for programming tasks through specific optimizations such as syntax-aware learning (Gong et al., 2024), specialized tokenization (Bavarian et al., 2022), and program translation (Du et al., 2025; Li et al., 2025).

**Code Generation with LLMs** CodeLLMs, such as StarCoder (Li et al., 2023c), StarCoder2 (Lozhkov et al., 2024), DeepSeek-Coder (Guo et al., 2024), and Qwen2.5-Coder (Hui et al., 2024) exhibit strong foundational programming capabilities and excel in code generation. To further enhance these models, Supervised Fine-Tuning(SFT) has been employed to improve task-specific performance and model alignment (Luo et al., 2024; Wei et al., 2023; Zheng et al., 2024). For example, Code Alpaca (Chaudhary, 2023) adopts the Self-Instruct method (Wang et al., 2023), a technique that utilizes ChatGPT to generate high-quality instruction datasets. In contrast, Vicuna (Peng et al., 2023) is fine-tuned on daily conversational datasets collected from a data-sharing platform. Similarly, WizardLM (Luo et al., 2024) utilizes the Evol-Instruct method to enhance instruction datasets iteratively, leading to more complex and diverse tasks. Meanwhile, SelfCodeAlign (Wei et al., 2024) employs a base model to infer aligned data. SFT for code generation is gaining significant attention (Zhu et al., 2024) as it enables models to adapt to specialized tasks and generate more precise and context-aware outputs.

**Chain of thought** The performance of LLMs heavily depends on the quality of input prompts (Gao et al., 2023b). Inspired by the success of Chain-of-Thought (CoT) techniques (Wei et al., 2022) in logical reasoning, researchers are exploring their applications in code generation. For example, (Li et al., 2023a) proposes a structured CoT approach to help models understand complex intentions and solve challenging problems. Jiang et al. (2023) introduces a self-planning method consisting of distinct planning and implementation phases. Additionally, Yang et al. (2024a) develops a technique that automatically uses lightweight language models to generate CoTs for code generation. Chen et al. (2023), Gao et al. (2023a), and Hu et al. (2023) use LLMs to output text and programming language statements, and finally, an answer. Current research primarily focuses on exploring various CoT prompting strategies and their underlying mechanisms (Madaan et al., 2023; Yang et al.,

2024d; Chen et al., 2024a; Kudo et al., 2024; Feng et al., 2023). Some recent studies have explored knowledge distillation of CoT from LLMs to smaller language models (Magister et al., 2023; Hsieh et al., 2023). However, these studies primarily focus on traditional logical tasks, paying limited attention to code generation.

## 3. Preliminaries

**Decoder-only LLM** In this work, we focus on a decoder-only large language model (LLM) as the base model for downstream tasks. Given an input sequence $x = (x_1, x_2, \ldots, x_l)$ of length $l$, the model processes $x$ through multiple transformer-based decode layers to produce hidden states $h$. The final hidden state $h_l$ is projected by a language modeling head Linear into logits $z$, which are normalized via Softmax function to yield the probability distribution $p$ over the vocabulary for the next token:

$$p = \text{Softmax}(z) \quad \text{and} \quad p_j = \frac{\exp(z_j)}{\sum_{k=1}^{|\mathcal{V}|} \exp(z_k)}$$

where $|\mathcal{V}|$ is the vocabulary size, and $z_j$ denotes the $j$-th logit. This mechanism enables LLM to generate the next token based on the input context, supporting various tasks.

**Supervised Fine-Tuning** Supervised Fine-Tuning (SFT) is essential for adapting LLM to specific applications. The SFT process can be described as follows: Given a labeled dataset $\mathcal{D} = \{(x^{(i)}, y^{(i)})\}_{i=1}^{|\mathcal{D}|}$, where $x^{(i)}$ is the input sequence and $y^{(i)}$ is the corresponding output with length $l_i$, the goal is to minimize the loss function $\mathcal{L}$ over the dataset:

$$\theta^* = \arg\min_\theta \sum_{i=1}^{|\mathcal{D}|} \sum_{t=1}^{|l_i|} \mathcal{L}\left(y_{t+1}^{(i)}, \text{Softmax}(\text{Linear}(h_t^{(i)}; \theta))\right)$$

where $\theta$ represents the model parameters, $h_t^{(i)}$ is the final hidden state for the context sequence $\left(x^{(i)} + y_t^{(i)}\right)$, and $y_{t+1}$ is the encoded token at timestamp $t$. The loss function $\mathcal{L}(y, p) = -\sum_{j=1}^{|\mathcal{V}|} y_j \log(p_j)$ is typically a cross-entropy loss, which measures the difference between the predicted probability distribution $p$ and the true one-hot label $y$.

**Pass@k** For evaluating CodeLLM performance, we mainly measure the likelihood that a model generates at least one correct code sample out of $k$ attempts for a given problem with the metric Pass@k $= 1 - (1 - p)^k$, where $p$ represents the probability of generating a correct sample, from (Chen et al., 2021; Kulal et al., 2019). To mitigate the computational challenges of directly calculating Pass@k, an unbiased estimation method is used:

$$\text{Pass@k} = 1 - \frac{\binom{n-c}{k}}{\binom{n}{k}} \tag{1}$$

where $\binom{n}{k}$ denotes the binomial coefficient, which represents the number of ways to choose $k$ items from $n$, while $\binom{n-c}{k}$ represents the number of ways to select $k$ incorrect samples from the $n - c$ incorrect samples. Under the setup of greedy decoding, Pass@1 corresponds to the proportion of problems in the benchmark successfully solved when the model generates a single deterministic output per problem with a temperature of 0.0.

## 4. Method

In existing open-source SFT datasets, CoT and Code components are commonly integrated into responses to code generation tasks. Inconsistent formats hinder the evaluation of different components' contributions to model performance. This section outlines our synthetic dataset to address these limitations. Our objective is to develop well-organized datasets designed to enable controlled experiments and analyze the impact of SFT on CodeLLM's performance.

### 4.1. Data collection

Previous studies often use code snippets from GitHub (Wei et al., 2023) or Common Crawl (Yang et al., 2024c) as seed data during the SFT stage of large code models, guiding the models in generating instruction-answer pairs. However, taking raw code snippets as seed data is often unlikely to create problems that are sufficiently challenging, diverse, and representative of instruction distributions (Song et al., 2024). Consequently, the solutions usually lack comprehensive reasoning steps and high-quality code solutions. To address this issue, we collect and filter program tasks from open-source datasets to construct a seed set, including Magicoder-OSS-Instruct (ISE-UIUC, 2024), the Python Code subset of ShareGPT (Innovations, 2023), Evol-CodeAlpaca (Tam, 2024), Evol-Instruct-Code (Roshdieh, 2023), CodeExercise-Python (CodeFuse-AI, 2023a), and Codefuse-Evol-Instruct (CodeFuse-AI, 2023b), as recommended by prior research. We put the detailed statistical information of these used datasets in Appendix C.

### 4.2. High-Quality CoT and Code

We start with an initial seed dataset consisting of program questions $x$ and their corresponding responses $y$, represented as $\mathcal{S} = \{(x_i, y_i)\}_{i=1}^{|\mathcal{S}|}$. As outlined in (Bai et al., 2022; Snell et al., 2022), we adopt the method of context distillation, which takes a general model $M_t$ as teacher to synthetic data, using it to SFT a base student model $M_s$. The teacher model can follow instructions to generate contextually relevant outputs and align them with specific behavioral objectives. It reduces the time and cost of labeling supervised data, requiring minimal human intervention.

Our technique starts with a small set $\mathcal{P}$ of examples where

$|\mathcal{P}| = 3 \ll |\mathcal{S}|$. By drawing inspiration from (Zelikman et al., 2022), we provide the original $(x_i, y_i)$ as the hint to prompt teacher model $M_t$, sample multiple times and then select the best CoT and Code to include in the few shots set $\hat{\mathcal{P}}$. Building on the In-Context Learning method proposed in (Huang et al., 2022), we use a few-shot prompt from $\hat{\mathcal{P}}$ and each seed from $\mathcal{S}$, calling $M_t$ to generate detailed reasoning steps $r_i$ and a high-quality code solution $c_i$ for $(x_i, y_i)$. The prompt template is in Appendix B. We assume incorrect reasoning or solutions may introduce noise into synthetic datasets, potentially hindering model learning. Therefore, we use the Self-Consistency (Wang et al., 2023) to drop part of the generated data, retaining only those that produce the correct answer. Specifically, we prompt the model $M_t$ to generate multiple test cases, selecting only the $c_i$ that successfully pass the test code. We use the most powerful open-source model currently available, DeepSeek-V2.5-1210 [1], as the teacher to construct our synthetic dataset. This process ensures that code snippets are syntactically valid and consistent, with reasoning standardized and aligned to the code content. The details can be found in Appendix A.

### 4.3. Different Strategies for SFT

We consider a base model $M_s$ with a filtered dataset $\mathcal{D} = (x_i, y_i, r_i, c_i)_{i=1}^{|\mathcal{D}|}$, and design four distinct data strategies and utilize them. During SFT, the model takes the problem description as input, with output varying by dataset format.

**Seed Dataset**   The initial dataset consists of questions $x$ and their corresponding response $y$, represented as $\{(x_i, y_i)\}_{i=1}^{|\mathcal{D}|}$. This dataset forms the foundation for training and can serve as the baseline, denoted as $Seed$.

**Code without CoT**   In the dataset $\{(x_i, c_i)\}_{i=1}^{|\mathcal{D}|}$ where CoT is excluded, the data format is more standardized as generated by the same model $M_t$. The model is specifically trained to output the Code, denoted as $C_{\text{w/o}}$.

**Code follow CoT**   In the dataset $\{(x_i, r_i + c_i)\}_{i=1}^{|\mathcal{D}|}$, the model is fine-tuned first to generate the CoT based on the problem, followed by the corresponding code solution. This approach adheres to the CoT prompting methodology commonly used in LLMs and is denoted as $C_{\text{follow}}$.

**Code precede CoT**   In contrast to the previous approach, the dataset $\{(x_i, c_i + r_i)\}_{i=1}^{|\mathcal{D}|}$ is used to fine-tune the model first to generate the code solution, followed by the corresponding CoT. This less conventional method takes CoT as an explanation and is denoted as $C_{\text{precede}}$.

---

[1] https://platform.deepseek.com

## 5. Experiments

We conduct experiments on several code generation benchmarks to evaluate the performance under different SFT strategies. The benchmarks used in our experiments are exhaustively described in Appendix D. We use OpenCodeEval (Liu, 2024) as our evaluation framework. Most experimental results are derived from training DeepSeek-Coder-Base-6.7B (Guo et al., 2024) for 3 epochs. The learning rate is dynamically adjusted using a cosine decay scheduler with a warm-up ratio of $0.1$ and a peak value of $1e-5$. The maximum sequence length is fixed at 4096 tokens during training. More information can be found in Appendix E.

### 5.1. Evaluation Results

We use the EvalPlus benchmark (Liu et al., 2023) to evaluate the code generation capabilities of fine-tuned models under different strategies. As shown in Table 1, $C_{\text{precede}}$ has 9.86% relative performance improvement compared with $C_{\text{follow}}$.

Table 1: The performance of different SFT models on code generation by HumanEval, MBPP, and extended versions.

| Method | HumanEval(+) | MBPP(+) | Average |
|---|---|---|---|
| $Seed$ | 68.29(62.20) | 77.51(65.61) | 68.40 |
| $C_{\text{w/o}}$ | 70.73(64.63) | 80.42(67.72) | 70.88 |
| $C_{\text{follow}}$ | 67.07(59.75) | 74.33(60.58) | 65.43 |
| $C_{\text{precede}}$ | 71.95(67.68) | 80.69(67.20) | 71.88 |

We further benchmark our SFT models using Live-CodeBench (Jain et al., 2024) to ensure a fair comparison. Notably, this benchmark enables evaluation of the models' out-of-distribution (OOD) generalization, including evolving test subsets reflecting temporal shifts in data. Table 2 presents Pass@1 results for subsets generated after different start dates to analyze temporal performance trends.

Table 2: The performance of different SFT models on Live-CodeBench. Newer Start Dates Reflect Lower Contamination for Evaluating OOD Generalization.

| Method | Contest Start date | | |
|---|---|---|---|
| | 2023-09-01 | 2023-07-01 | 2023-05-01 |
| $Seed$ | 20.12 | 21.77 | 23.89 |
| $C_{\text{w/o}}$ | 20.47 | 21.67 | 23.51 |
| $C_{\text{follow}}$ | 20.69 | 22.31 | 23.07 |
| $C_{\text{precede}}$ | 21.08 | 22.56 | 24.16 |

To deeply understand the advanced capabilities of our fine-tuned model, we extend our evaluation by using Big-CodeBench (Zhuo et al., 2024). This allows us to specifically examine its performance in two crucial areas: following complex natural language instructions and calling

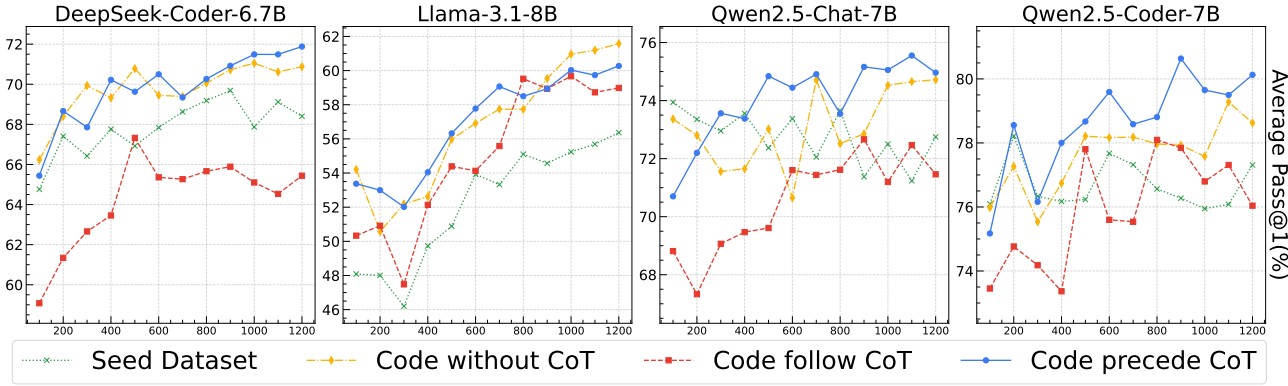

Figure 2: The average performance of various base models fine-tuned under different strategies across steps (x-axis).

external function tools. As shown in Table 3, the primary results are explicitly derived from the benchmark's instruction split, isolating the model's ability to interpret and execute given instructions accurately.

Table 3: The performance of different SFT models on Big-CodeBench. Different task results are obtained for the Full set and a more challenging Hard subset.

| Method | Task Set | |
|---|---|---|
| | Full | Hard |
| $Seed$ | 37.71 | 13.51 |
| $C_{\text{w/o}}$ | 38.25 | 15.54 |
| $C_{\text{follow}}$ | 36.05 | 15.54 |
| $C_{\text{precede}}$ | 38.33 | 16.22 |

We place the other results in Appendix F and significance analysis in Appendix G. Our experimental results show that teaching the model to write CoT before code does not benefit downstream tasks, but if we treat CoT as an explanation of the code, the model performs better. The classic CoT training method, which typically organizes reasoning steps before generating a solution, does not work well on code generation tasks. The code is a reasoning process, while the latter CoT helps to understand. These findings highlight the importance of the CoT position and verify that the code itself is a reasoning process.

### 5.2. Study of SFT Configurations

This study examines whether our conclusions hold across different pre-training datasets and model architectures by varying the base models. The choice of the base model is critical, as it determines the initial capabilities and foundational architecture of language models, which in turn affects downstream task performance. The experiment involves fine-tuning various base models of different sizes and architectures while keeping all other hyperparameters constant.

Model performance is evaluated at checkpoints every 100 training steps. The results on EvalPlus, as shown in Figure 2, reveal that the base model has a notable effect on SFT outcomes; this observation aligns with findings widely reported in the literature. Although a distributional gap may inevitably exist between our high-quality SFT data and the pre-training data, different models still benefit from explanation after coding and avoiding overthinking. We also discuss other key hyperparameters of SFT in Appendix H.

### 5.3. Study of Teacher Model

To examine whether our conclusions remain valid when synthetic data is generated using different teacher models, we employ GPT-4o-0806 as an alternative teacher model to regenerate the dataset to validate our approach further. We then fine-tuned the same student model using the methodologies outlined above. The models are subsequently fine-tuned and evaluated under the same settings described above. The results, detailed in Table 6, demonstrate a consistent trend.

Table 4: The performance of different SFT models on the EvalPlus benchmark. These models are fine-tuned on another dataset synthesized by GPT-4o-0806.

| Method | HumanEval(+) | MBPP(+) | Average |
|---|---|---|---|
| $C_{\text{w/o}}$ | 71.95(65.85) | 77.77(67.46) | 70.76 |
| $C_{\text{follow}}$ | 65.24(59.14) | 76.45(62.96) | 65.95 |
| $C_{\text{precede}}$ | 72.56(66.46) | 78.57(67.72) | 71.33 |

### 5.4. Study of Synthesis Order

To mitigate the bias introduced by providing reference code during data synthesis, we modify our data generation approach to investigate the synthetic order's impact and ensure that the teacher model must generate CoT reasoning without access to code-related information. Instead of providing a reference solution, we directly utilize educational instructions from the SelfCodeAlign (Wei et al., 2024) dataset to

synthesize data. The results are listed in Table 5.

Table 5: The performance of different SFT models on the EvalPlus benchmark. These models are fine-tuned on another dataset synthesized by a different order.

| Method | HumanEval(+) | MBPP(+) | Average |
|---|---|---|---|
| $C_{\text{w/o}}$ | 67.07(60.97) | 76.71(60.31) | 66.27 |
| $C_{\text{follow}}$ | 62.19(54.87) | 71.95(57.93) | 61.74 |
| $C_{\text{precede}}$ | 68.29(60.36) | 78.57(62.16) | 67.35 |

### 5.5. Study of Data Source

To investigate whether our conclusions can be applied to the pre-training stage, we utilize Stack (Kocetkov et al., 2022) to synthesize data as an additional data source, which is a 3.1 TB dataset of permissively licensed source code in 30 programming languages, designed to foster open and responsible research on CodeLLMs. Subsequently, the models are trained and tested following the previously outlined setup. Table 6's results reveal a clear and consistent pattern.

Table 6: The performance of different SFT models on the EvalPlus benchmark. These models are fine-tuned on another dataset synthesized from the Stack's codes.

| Method | HumanEval(+) | MBPP(+) | Average |
|---|---|---|---|
| $C_{\text{w/o}}$ | 68.29(61.58) | 76.98(62.43) | 67.32 |
| $C_{\text{follow}}$ | 61.58(55.48) | 75.13(60.31) | 63.13 |
| $C_{\text{precede}}$ | 69.51(64.63) | 77.24(63.22) | 68.65 |

## 6. Discussions

This section analyzes several key aspects of model behavior under different strategies: **1**. The factors influencing performance variations when changing the position of CoT in the training. **2**. The specific data components that enhance the model's capability in code generation tasks during the SFT process. **3**. The generalizability of our proposed conclusions across different scenarios. **4**. The distinctive characteristics of outputs generated by various SFT models.

### 6.1. Model Behavior Analysis

**Conditional Perplexity Gap** In our experiments, we analyze how the order of information affects the difficulty of learning data (Li et al., 2024c;b) by examining the distributions of perplexity across two strategies: $C_{\text{precede}}$ and $C_{\text{follow}}$. We visualize the perplexity of the preceding part in both setups and analyze the conditional perplexity of the latter part, considering the former as its context. As shown in Figure 3, while the overall perplexity distributions are similar, placing the Code first reduces the gap between the two distributions. The result suggests that the model can better

balance the learning in the two parts. These differences arise because source code has stricter syntax and semantics than CoT. Therefore, some tokens are easily inferred based on grammatical rules when generating Code. In contrast, CoT permits greater flexibility and may cause overthinking behavior (Chen et al., 2024b) when generated first.

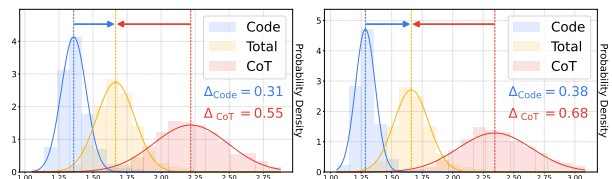

Figure 3: Conditional PPL Distribution Histogram with Gaussian Fit: $C_{\text{precede}}$ (left) and $C_{\text{follow}}$ (right), representing learning difficulty gap on different parts in the dataset.

**Memorization and Generalization** To further investigate the impact of different orders of context on learning efficiency, we analyze the model's memorization and generalization before and after training. Figure 4 compares two SFT strategies by evaluating them on the validation set. The left plot depicts the KL divergence between the fine-tuned models (after training) and the base model (before training) under two distinct data strategies. The minor difference in KL divergence suggests that both SFT strategies modify the base model similarly. Conversely, the right plot shows the validation loss, reflecting the accuracy of the next-token predictions. The notable difference in this plot shows that the two SFT strategies leverage the data in distinct ways. The results suggest that while both strategies fit the training set similarly, the $C_{\text{precede}}$ strategy exhibits significantly better generalization, as reflected in code generation ability.

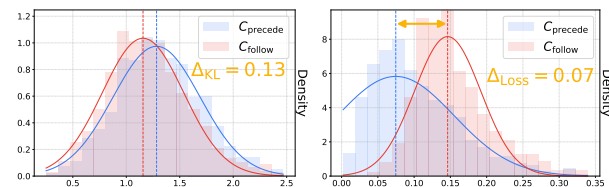

Figure 4: The Distribution Histogram Plot with Gaussian Fit lines: the KL Divergence between Base and SFT Models (left) and the Validation Loss (right).

**Context Attention Weight** In our experiments, we investigate the effect of information ordering on the model's attention weights, specifically examining the influence of natural language instructions, CoT, and Code. We apply bilinear interpolation to downsample the average attention weight matrix from the model's final layer to a fixed spatial resolution, ensuring consistent comparison across inputs with different orderings. The visualization in Figure 5 indi-

cates that Code exhibits no distinct attention bias towards CoT, with attention values evenly distributed across the regions corresponding to Code segments. However, when Code precedes CoT, we observe that the model allocates increased attention to Code, potentially indicating an attempt to better comprehend its relationship with CoT. These results suggest that employing the $C_{\text{follow}}$ strategy during SFT alters the model's expectations regarding the underlying data distribution. Consequently, enforcing a fixed attention order between CoT and Code might hinder the model's ability to adapt to these constraints. This misalignment could subsequently impair the model's generalization capabilities on downstream tasks.

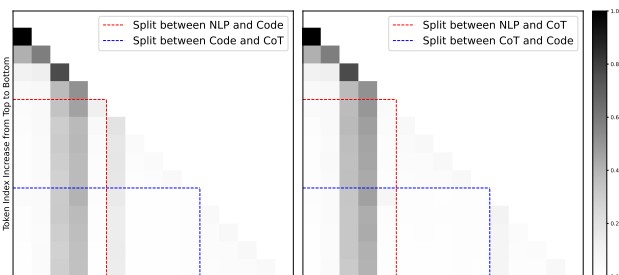

Figure 5: The Visualization Results of Compressed Attention Weight Matrices in the last layer under Different Strategies: $C_{\text{precede}}$ (left) and $C_{\text{follow}}$ (right).

**Layer Gradient Norm** In this subsection, we aim to analyze the gradients in the dynamic training process of LLMs, particularly their scaling and distribution across layers. We investigate the gradient across different LLM layers under various data parts while maintaining the same initial model state. We primarily analyze the gradients of attention-related layers, including $Q_{\text{Proj}}$, $K_{\text{Proj}}$, $V_{\text{Proj}}$, and $O_{\text{Proj}}$. Given the immense number of layers and parameters in current LLMs, it is computationally infeasible to analyze these large gradient matrices directly. Therefore, we use the $\ell_2$ norm to capture the gradient characteristics of each layer, particularly its strength. Figure 6 shows that different data have different impacts on the parameters, and as we mentioned earlier, the overall impact of the code part is relatively weak. These findings highlight the significant effect of training order on gradient contributions.

### 6.2. Data Pattern Discussion

**Inconsistent Data Analysis** In our experiments, we analyze the impact of inconsistent Code, defined as Code that fails to pass its self-generated test cases, on model performance under various strategies. As depicted in the left plot of Figure 7, the results indicate that conclusions from our previous study on various data strategies remain valid; the model exhibits only a marginal decrease in performance

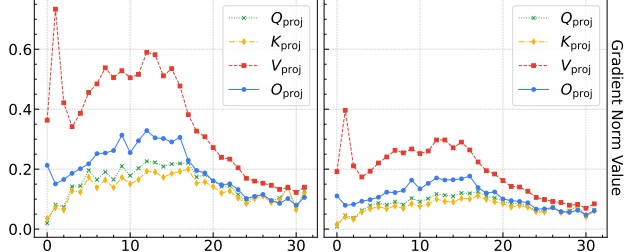

Figure 6: The $\ell_2$ norm of gradients across different layers (x-axis) under two parts: CoT (left) and Code (right).

when compared to models trained on self-consistent data. These results suggest that consistency is not the sole determinant of code quality, particularly given that LLMs often exhibit limited proficiency in generating accurate test cases. Even Code containing errors can offer valuable insights and direction for the model's learning process. These findings underscore that high-quality Code, even if imperfect, accompanied by a detailed generation process, serves as a valuable training signal for CodeLLMs.

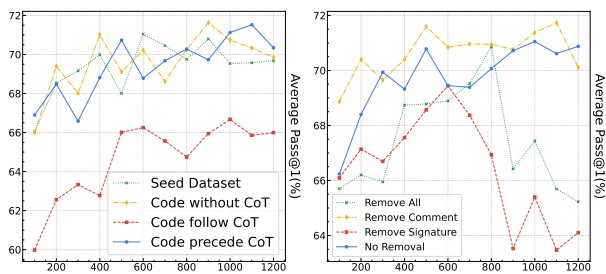

Figure 7: The average performance on the EvalPlus benchmark of two studies: inconsistent data study (left) and signature removal study (right).

**Signature Removal Analysis** Our experiments in this subsection investigate whether annotation components within the Code significantly affect the model's understanding and learning capabilities. Specifically, we examine two forms of annotations: function or class signatures and inline code comments. We analyze how removing these annotation elements from the dataset impacts model performance across various benchmarks. The right plot of Figure 7 illustrates that removing signatures leads to the most significant performance decline throughout training, highlighting the critical role of signatures in facilitating the model's learning process. These findings suggest that function or class signatures are crucial in code generation, as they bridge the gap between natural language and programming languages (Yang et al., 2024b). Conversely, removing inline comments does not appear to have a significant impact, likely due to different code styles of LLMs.

**Mixed Pattern Analysis** This study examines whether LLMs can benefit from a mixed-data strategy, specifically investigating whether diverse data distributions enhance model generalization. To this end, we combine the $C_{follow}$ and $C_{precede}$ strategies and fine-tune the base model for one epoch. For a fair comparison, we apply these strategies separately for two epochs, ensuring an equivalent number of training steps. The results presented in the left plot of Figure 8 reveal that the mixed-data strategy yields lower performance than a single strategy. The experiment suggests that the mixed-data strategy may be more challenging for the model to learn, implying an increase in the overall difficulty of the dataset from the model's perspective. Presenting an LLM with two different answers for the same problem establishes two distinct optimization objectives.

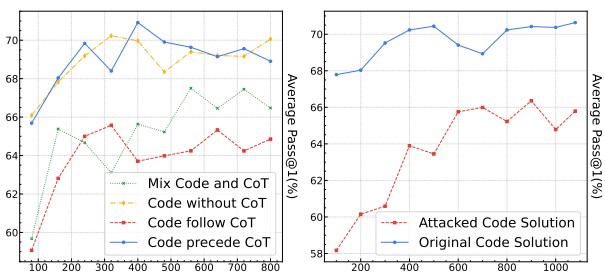

Figure 8: The average performance on the EvalPlus benchmark of two studies: mixed pattern dataset (left) and attacked code dataset (right).

**Adversarial Code Robustness** Previous studies suggest that LLMs learn programming languages by processing Code as sequences of tokens and are easy to attack due to their strong dependence on lexical features. Therefore, we design a lexical-level attack on code snippets that preserves syntax and semantics to investigate the impact of training a model on manipulated data. Specifically, we replace all variable names in the Code with random, meaningless strings, thereby limiting the model's access to lexical information. The right plot of Figure 8 illustrates an assessment of model robustness under adversarial conditions by comparing performance on original solutions with that on the attacked Code. Although an upward trend is observed with increasing training steps, performance on the attacked solutions remains consistently lower, suggesting that the model has difficulty mitigating the effects of adversarial noise, a finding consistent with (Yan & Li, 2021).

### 6.3. Generalization Discussion

**Generalization Across Sizes** To assess the robustness of our findings, we extend the experimental setup to include two larger language models: DeepSeek-Coder-33B-Base and Qwen2.5-Coder-32B-Base. The results presented in Figure 9 demonstrate that the conclusions drawn from our

initial analysis remain consistent across these larger architectures. These findings highlight the scalability and general applicability of the insights derived from our experiments, thereby reinforcing the importance of thoughtful dataset construction for maximizing performance in large-scale models.

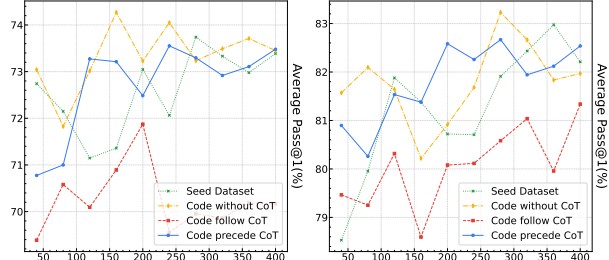

Figure 9: The average performance on the EvalPlus benchmark of large size: DeepSeek-Coder-33B-Base (left) and Qwen2.5-Coder-32B-Base (right).

**Generalization Across Difficulties** To explore whether the $C_{precede}$ approach generalizes effectively to more complex problems, we design an experiment to assess the robustness of this data strategy in challenging scenarios. Table 7 reveals that incorporating CoT explanation consistently improves model performance, particularly for tasks requiring higher logical complexity. Furthermore, the relative positioning of the CoT part demonstrates a measurable impact on outcomes, underscoring the importance of data strategy. These findings suggest that considering the CoT position into SFT is effective and exhibits strong potential for scalability to more complex problems. Our method offers a robust framework for enhancing reasoning in code generation tasks.

Table 7: The performance of different strategies on the LeetCode Contest Benchmark with three difficulty levels.

| Method | Easy(45) | Medium(91) | Hard(44) | Overall(180) |
|---|---|---|---|---|
| $Seed$ | 42.2%(19) | 15.4%(14) | 9.1%(4) | 20.6%(37) |
| $C_{w/o}$ | 42.2%(19) | 16.5%(15) | 2.3%(1) | 19.4%(35) |
| $C_{follow}$ | 31.1%(14) | 14.3%(13) | 4.5%(2) | 16.1%(29) |
| $C_{precede}$ | 44.4%(20) | 24.2%(22) | 6.8%(3) | 25.0%(45) |

**Discuss Pass@k Impact** Different $k$ values reflect different aspects of model performance, which reveals different evaluations of model behavior and potential. Figure 10 left illustrates the effect of varying $k$ in Eq. 1 on the average performance across different benchmarks. As $k$ increases, all strategies demonstrate an improvement in accuracy, indicating models exploring more possibilities and increasing the chances of producing correct outputs. The results show that $C_{precede}$ almost outperforms all other strategies, highlighting the importance of positioning the CoT explanations.

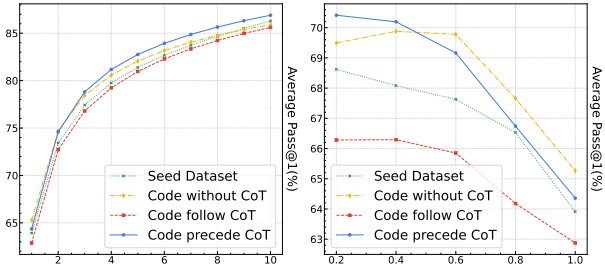

Figure 10: The average performance on the EvalPlus benchmark of different $k$ in Pass@k (left) and temperature (right).

**Discuss Temperature Impact**   When LLMs generate the answers, the temperature affects the model's performance, with low temperatures tending to make the model's output more fixed and high temperatures encouraging more exploration in action. In the right plot of Figure 10, we aim to show the effect of varying the temperature parameter. As the temperature increases, the overall performance tends to decline across all strategies, with the steepest drop observed in $C_{\text{precede}}$. The result highlights the trade-off between diversity and accuracy, as higher temperatures lead to more diverse but less precise outputs. $C_{\text{precede}}$ maintains a relatively stable performance, reinforcing its robustness.

### 6.4. Model Outputs Discussion

**LLM-as-a-Judge**   To address the high cost of collecting human preferences (Chiang et al., 2024), we adopt a pairwise comparison approach, where an LLM evaluates a question alongside two answers and determines which response is of higher quality. We prompt GPT-4o-0806 to assess different parts of the responses based on specific criteria. Each row represents a different evaluation part, and each column represents the winner, the response the evaluator thinks is better. We change the response order and conduct evaluations twice to enhance effectiveness. The results in Table 8 indicate that the $C_{\text{precede}}$ approach aligns more closely with real-world preference on all three parts.

Table 8: Result of LLM-as-a-Judge of different strategies.

| Winner | $C_{\text{follow}}$ | Tie | $C_{\text{precede}}$ |
|---|---|---|---|
| CoT | 22.1%(242) | 52.8%(577) | 25.1%(274) |
| Code | 20.1%(228) | 51.0%(557) | 28.2%(308) |
| Total | 23.2%(254) | 51.2%(560) | 25.5%(279) |

**Length and Quality**   The study presents a comparative analysis of the outputs generated by $C_{\text{follow}}$ and $C_{\text{precede}}$ concerning the CoT and the Code in outputs. Besides the number of tokens and steps, we report the BLEU score for CoT, which measures alignment with the teacher output

in Table 9. Similarly, we report the CodeBLEU score for code outputs to assess their syntactic and semantic similarity to the expected Code. The results indicate minimal differences between the two models in terms of BLEU. However, $C_{\text{precede}}$ can achieve a marginally higher CodeBLEU score for the code outputs with fewer steps and tokens, which means it gets better results with less inference cost.

Table 9: Comparison of CoT and Code parts' length and quality under different SFT strategies.

| Method | CoT | | | Code | | |
|---|---|---|---|---|---|---|
| | Steps | Tokens | BLEU | Steps | Tokens | CodeBLEU |
| $C_{\text{follow}}$ | 5.05 | 197.99 | 60.34 | 13.02 | 258.94 | 68.19 |
| $C_{\text{precede}}$ | 5.02 | 195.46 | 60.22 | 12.02 | 256.97 | 68.96 |

**Response Match Instruction**   We analyze the similarity between the instruction and two other components: the Code and the CoT. Specifically, we use OpenAI's embedding model text-embedding-3-large to obtain the representation of the text and then compute the cosine similarity between them. This evaluation assesses the alignment and coherence between the instructions and the corresponding outputs, including the executable Code and the reasoning processes in the CoT. By comparing these similarities in Figure 11, we aim to gain insight into the degree of semantic match correspondence and the extent to which the generated outputs adhere to the given instructions.

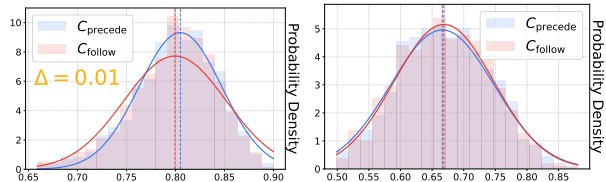

Figure 11: Cosine Similarity Distribution with Gaussian Fit: Instruction and Code (left), Instruction and CoT (right).

## 7. Conclusion

In this study, we investigate the impact of incorporating CoT reasoning into the SFT process to improve code generation in LLMs. By constructing a high-quality dataset, we facilitate comprehensive experimentation across a range of base models and configurations. Our results show that high-quality code can already serve as a reasoning process, while traditional CoT should be regarded as an explanation of the code. Additionally, we gain key insights into how CoT strategies and data patterns influence fine-tuning outcomes, providing practical guidelines for future studies. These findings highlight the effectiveness and scalability of CoT strategies in SFT for the code generation model.

## Acknowledgements

This paper is supported by NSFC (62076121) and the Major Program (JD) of Hubei Province (2023BAA024). The authors thank Dr. Wang-Zhou Dai and Hao-Yuan He for their insightful discussions regarding this work.

## Impact Statement

Our work provides critical insights into how LLMs learn to reason through CoT in code generation, challenging conventional training paradigms. Beyond immediate performance gains, we offer novel insights into how LLMs learn to reason with CoT data for code generation, revealing that restructuring the order significantly improves performance. Furthermore, our exploration of the properties of CoT training and data strategies contributes to advancing reasoning methodologies in LLMs, potentially shaping the development of effective and scalable AI-driven software engineering.

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

# Appendix

## A. Synthetic Dataset Process

To select suitable seed data, we established specific rules to choose questions that have educational value and are worth reasoning. After that, we use AST to parse the Python code in the answer, ensuring that we do not provide answers with errors that would prompt the model to reject them. The overall process is illustrated in Figure 12.

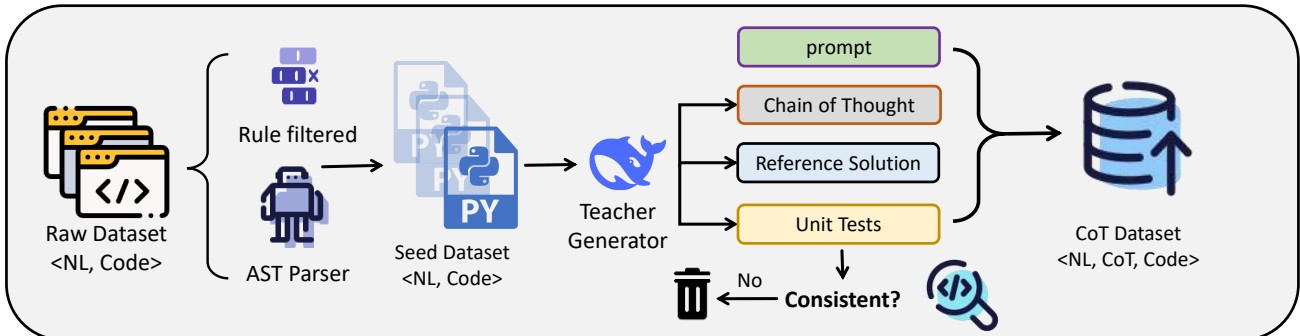

Figure 12: Synthetic Dataset Process

## B. Prompts Template

Here is the prompt for synthetic self-consistent code generation:

---

**Prompt for Instruction Synthesis**

You are a teaching assistant helping to create a Python programming task from a given code snippet. You will respond best to the Python programming task, including the reasoning process, reference solutions, and test code.

**[Code Snippet]**
{Code}

The response must have these parts:
**[Analysis]**
{Analyze the task and reason about the given task step by step}
**[Solution]**
{Write a high-quality reference solution in a self-contained script that solves the task}
**[Test]**
{Provide ten assert statements to check the correctness of your solution}

---

## C. Dataset statics

In this section, we present the details of the datasets in Table 10. Finally, we obtain a comprehensive dataset comprising 52,293 samples. We randomly select 51,200 as the training set and the rest as the validation set.

## D. Benchmark

**HumanEval**   The HumanEval dataset(Chen et al., 2021), proposed by OpenAI, comprises 164 handwritten programming problems that cover areas such as algorithms, mathematical computations, and string manipulation. Each programming problem includes a natural language description of the requirements, a function signature, and test cases.

**Table 10:** Dataset statics

| Dataset | Total | Filtered | Self-Consistent | Ratio |
|---------|-------|----------|-----------------|-------|
| CodeExercise-Python | 27,224 | 147,36 | 6,963 | 27.56% |
| Codefuse-Evol-Instruct | 66,862 | 16,176 | 8,000 | 11.96% |
| Evol-Instruct-Code | 78,264 | 23,175 | 12,003 | 15.34% |
| Evol-CodeAlpaca | 111,272 | 27,566 | 13,529 | 12.16% |
| Python ShareGPT | 22,608 | 14,061 | 6,131 | 27.11% |
| Magicoder-OSS-Instruct | 75,197 | 14,499 | 5,667 | 7.54% |

**MBPP** MBPP (Mostly Basic Programming Problems) (Austin et al., 2021) is a benchmark dataset proposed by Google. It consists of 974 programming tasks written by humans and reviewed by experts, involving simple numeric manipulations or basic usage of standard libraries.

**HumanEval+ and MBPP+** HumanEval+ and MBPP+ (Liu et al., 2023) are advanced benchmarks designed to provide more challenging evaluations for CodeLLMs. They combine LLM-based and mutation-based input generation to provide diverse test inputs for accurately evaluating the correctness of LLM-generated code. These two benchmarks provide a more comprehensive assessment of a model's coding capabilities, enabling a more thorough examination of CodeLLMs.

**LiveCodeBench** LiveCodeBench (Jain et al., 2024) is a contamination-free benchmark for evaluating code generation models. These tasks are curated from online judge websites, such as Codeforces and LeetCode, and feature an average of over 20 test cases per task. Although LiveCodeBench is a comprehensive benchmark addressing four problem types, the present study focuses on the code generation task to assess the function generation capabilities of LLMs.

**BigCodeBench** BigCodeBench (Zhuo et al., 2024) is a benchmark that challenges LLMs to invoke multiple functions as tools across 1,140 fine-grained tasks spanning 139 libraries and seven domains. In addition, the benchmark introduces a variant, BigCodeBench-Instruct, which automatically converts original docstrings into concise instructions. BigCodeBench also features a subset named BigCodeBench-Hard, which comprises 148 challenging programming tasks.

**MultiPL-E** MultiPL-E (Cassano et al., 2022) serves as a benchmark to evaluate the effectiveness of CodeLLMs in programming languages other than Python. MultiPL-E extends unit test-based evaluations to numerous languages by translating two established Python benchmarks into 18 different programming languages. This diverse set of languages enables a detailed analysis of how language frequency and structural features impact model performance.

**EvalPerf** EvalPerf (Liu et al., 2024a) demonstrates the effectiveness of the Differential Performance Evaluation (DPE) framework and provides valuable insights into program efficiency. The DPE framework addresses these limitations by focusing on computationally intensive tasks and implementing more stringent performance evaluation criteria. It provides a structured approach for evaluating the code generation efficiency of LLM.

# E. Experiment setup

The experiments are conducted on eight NVIDIA A100-SXM4-80GB GPUs, utilizing mixed-precision training (BF16) to enhance computational efficiency and reduce memory usage. We use ZeRO-3 (Rajbhandari et al., 2020) from Deep-Speed (Rasley et al., 2020) to minimize memory consumption during training. In our experiments, the micro-batch size is fixed at 4 per GPU to ensure consistent data processing across devices. At the same time, the gradient accumulation steps are varied based on the training configurations. To accelerate the training process, we employ FlashAttention-2 (Dao et al., 2022), a memory-efficient attention mechanism, to be the implementation for LLMs. The optimizer used was AdamW.

We monitored performance trends on specified benchmarks during training to systematically evaluate fine-tuned CodeLLMs under different SFT strategies. The models are initialized using a variety of pre-trained base models. To ensure the robustness and consistency of our findings, we systematically studied key hyperparameters, including batch size, learning rate, and total epochs, under controlled experimental conditions. In various experiments, we adjusted the save step during training to ensure a comparable number of checkpoints were available to monitor model performance trends.

## F. Benchmark results

We use EvalPerf to evaluate the efficiency of code generated by LLMs. Since efficiency is only essential when the generated code is correct, we focus our analysis on models with a pass rate of over 50%. Specifically, we utilize default settings, where each model generates 100 samples for each task at a temperature of 1.0. For tasks that produce at least 10 correct outputs, we analyze up to 20 correct samples from each model. Model rankings are based on win rates, with pairwise comparisons conducted using the DPS method on a standardized set of valid tasks. The result is shown in Table 11.

**Table 11:** EvalPerf results.

| Method | DPS | Pass@1 | Task Win Rate | Model Win Rate |
|---|---|---|---|---|
| $Seed$ | 81.8% | 69.4% | 45.1% | 40.5% |
| $C_{\text{w/o}}$ | 82.5% | 67.8% | 46.2% | 45.9% |
| $C_{\text{follow}}$ | 80.8% | 70.9% | 44.0% | 16.2% |
| $C_{\text{precede}}$ | 84.0% | 69.8% | 50.2% | 70.3% |

To explore whether the knowledge acquired from Python can be transferred to other programming languages, we compare the pass@1 accuracy of different fine-tuned models on MultiPL-E. Results in Table 12 show that the model series have different multilingual coding proficiency. Moreover, we find that the reasoning abilities the model acquired from training on synthetic Python datasets can be transferred to other programming languages, and our conclusion remains valid across different programming languages.

**Table 12:** MultiPL-E. results.

| Method | C++ | Java | PHP | Bash | Average |
|---|---|---|---|---|---|
| $Seed$ | 54.04% | 51.90% | 50.93% | 26.58% | 45.86% |
| $C_{\text{w/o}}$ | 54.04% | 52.53% | 52.17% | 32.28% | 47.76% |
| $C_{\text{follow}}$ | 53.42% | 52.53% | 51.55% | 31.65% | 47.29% |
| $C_{\text{precede}}$ | 54.66% | 54.43% | 56.52% | 34.81% | 50.11% |

## G. Significance Analysis

We evaluate the model's performance on the EvalPlus benchmark and use a t-test to show the statistical significance. We re-run five experiments on the synthetic dataset with different random seeds and fine-tune the DeepSeek-Coder-6.7B-Base model. The results in Table 13 show that $C_{\text{precede}}$ has a significant performance improvement over the $C_{\text{follow}}$ strategy.

**Table 13:** Significance Analysis Results.

| Random Seed | 0 | 1 | 2 | 3 | 4 | Average |
|---|---|---|---|---|---|---|
| $C_{\text{w/o}}$ | 69.19 | 70.06 | 70.34 | 70.48 | 70.83 | 70.17±0.62 |
| $C_{\text{follow}}$ | 64.88 | 66.54 | 66.03 | 66.38 | 65.84 | 65.93±0.65 |
| $C_{\text{precede}}$ | 71.49 | 71.18 | 71.82 | 71.14 | 70.96 | 71.31±0.34 |

## H. SFT Configurations Study

**Study of Batch Size**   This study investigated whether varying the batch size during fine-tuning significantly affects the change in model performance, which in turn influences computational efficiency and memory usage. This experimental setup aims to enhance the adaptability of the conclusion to varying batch sizes, thereby supporting training scenarios with limited GPUs. Model performance was assessed at checkpoints occurring every 10% of the training steps to enable fair comparisons across batch sizes. The results, shown in Figure 13, indicate that varying batch sizes do not affect our findings.

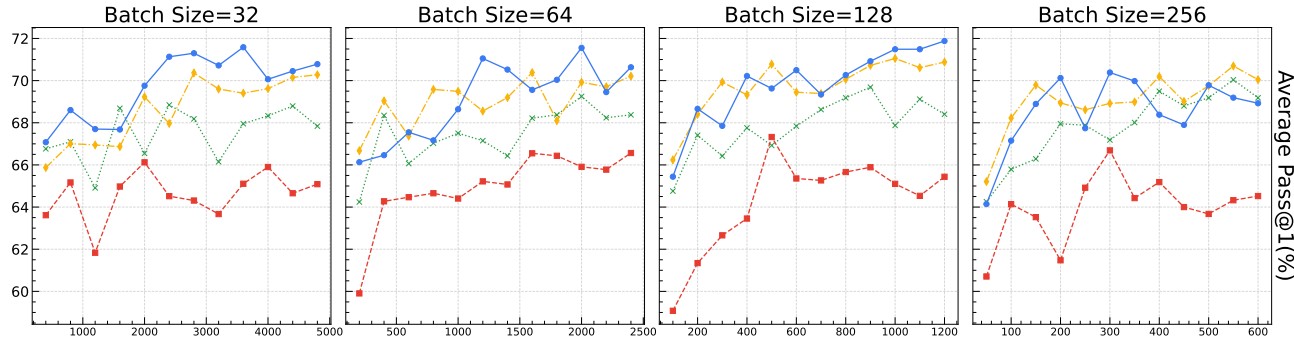

Figure 13: Average Performance of base models fine-tuned under different batch sizes across four benchmarks.

**Study of Train Epoch**     This study investigated whether varying the number of training epochs affects the conclusion in SFT. Previous work (Hoffer et al., 2017) has shown that the epoch count is a crucial parameter in closing the generalization gap during large-batch training of neural networks, including LLMs. It determines the number of dataset passes and significantly influences memorization and generalization. The performance of the SFT checkpoints is evaluated at every 10% of the training steps to ensure comparability across configurations. The results in Figure 14 can also verify our findings.

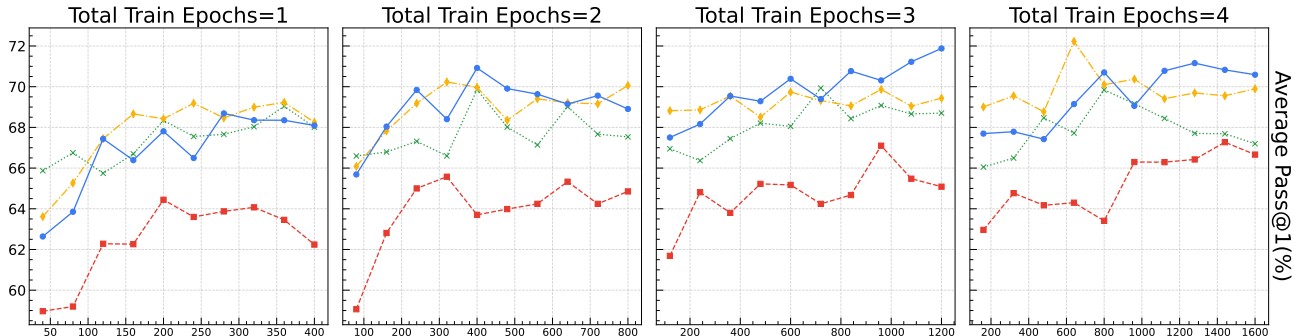

Figure 14: Average Performance of base models fine-tuned under different total epochs across four benchmarks.

**Study of Learning Rate**     This study investigated whether varying the learning rate during fine-tuning has a significant impact on model performance change. The learning rate is crucial for the speed and stability of convergence, both of which are critical for optimizing LLMs (Defazio et al., 2023; Abuduweili & Liu, 2024). Model performance is assessed at checkpoints spaced every 10% of the total training steps to ensure fair comparisons across learning rates. The results, shown in Figure 15, indicate that variations in learning rate do not change the validity of our conclusions.

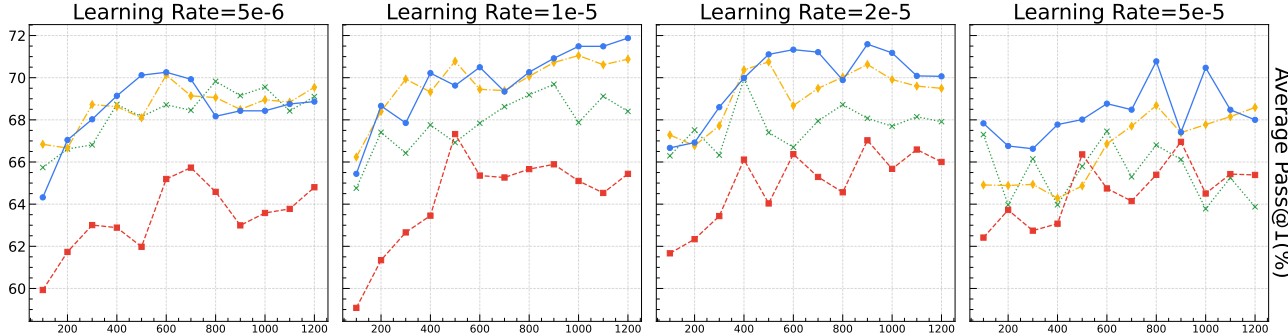

Figure 15: Average Performance of base models fine-tuned under different learning rates across four benchmarks.

