# OpenReview forum: "Revisiting Chain-of-Thought in Code Generation: Do Language Models Need to Learn Reasoning before Coding?"
_ICML.cc/2025/Conference — ICML 2025 poster_

### Official Review · Reviewer_793T · 2025-02-16

**Overall Recommendation:** 2

**Summary:**

The paper explores the impact of the data formatting when finetuning LLMs with synthetically generated data. Specifically, the data--composed of code and corresponding reasoning steps for a NL-to-code problem--is generated by a stronger "teacher" model (the authors state that it is a DeepSeek model in Appendix B, but do not specify which one). The data is then formatted in different ways by either putting the reasoning before or after the code snippet, or leaving it out altogether. The authors find that, somewhat surprisingly, including the reasoning before the code snippet does not seem to benefit downstream performance. They then go on to argue that including it *after* the code snippet may in fact benefit the SFT'd model the most.

## update after rebuttal
I thank the authors for a productive and engaging rebuttal discussion, which has certainly strengthened the findings of the paper. In particular, the addition of proper statistical testing, which should have been included in the original paper given the many possible confounding factors present, was much appreciated.

In the end, I have decided to nonetheless keep my original score of 2 (Weak Reject). If this is surprising to the authors then suffice it to say that I sympathize with them regarding the difficulty of presenting convincing evidence for counterintuitive findings. In its current form, the paper has me convinced that there is something amiss in the way we think about combining reasoning and code, but what exactly that is I am not convinced of. I note that the authors surely have tried to answer this question from a bunch of different angles, but none of them seem to have shed much light on the issue; in particular, none of the experiments in Section 6.1 seem quite insightful enough to me.

I will note to the AC that my vote for rejection is indeed to be taken to be quite weak; if the other reviewers are happy for the paper to be accepted in its current form then I will not vehemently oppose it. I simply think that the authors seem to have struck upon something deeply surprising and interesting, and with a few more months would likely be able to uncover what the cause is in much greater and more illuminating detail, ultimately leading to a much stronger contribution to the literature. As is, the paper will simply raise a few eyebrows, but not offer any lasting insights (in my opinion).

**Claims And Evidence:**

I do not think the claim/observation that formatting the data such that the reasoning follows the code benefits performance is supported by the evidence. This findings appears a bit hand-picked; there are many experiments in which, for example, Code w/o CoT does better than $C_\text{precede}$, but this is not even acknowledged (Fig 10, right, high temps; Fig 9, left; Fig 7, right; Fig 2, llama).

The large number of completely unrelated experiments, many of which are then not even discussed much in the text, does not inspire a great deal of confidence in the results being statistically significant, either. For example, the discussion of the last experiment is simply "we aim to gain insight into the degree of semantic match correspondence and the extent to which the generated outputs adhere to the given instructions." That just says what you *wanted* the experiment to tell you; it says absolutely nothing about what you then actually found, and indeed the figure does not seem particularly telling either.

**Essential References Not Discussed:**

I was surprised that Austin et al's "Program Synthesis with Large Language Models" (2021) was not cited as an example of early use of LLMs for code, since it is typically looked at as the seminal paper in the field.

More importantly, there does not appear to be a citation for pass@k, even though the unbiased estimate employed by the authors was first introduced in the Codex paper ("Evaluating Large Language Models Trained on Code", Chen et al., 2021). I believe the original idea of a pass@k metric is due to Kulal et al.'s "SPoC: Search-based Pseudocode to Code" (2019).

**Experimental Designs Or Analyses:**

Evaluating different data formats for SFT by simply doing SFT with a few different base models makes sense. This is the core experiment in the paper, which I am on board with.

However, there are many, many "side experiments" that I do not really understand the relevance of. Take for example the data mixing experiment: training on data is inconsistently formatted hurts performance. Is that really a finding? What does that tell you about how to format your data? To my eyes, this is a complete non-result.

The worst case of this at the end of P.7, when the authors "evaluate how varying the number of samples `k` in `pass@k` affects performance for the different methods) (Figure 10, left), and conclude that: "As k increases, all configurations demonstrate an improvement in accuracy, indicating models exploring more possibilities and increasing the chances of producing correct outputs". This doesn't even make any sense as an experiment; pass@k as a metric is monotonically non-decreasing by construction, since if you draw more samples i.i.d. the probability that at least one of them is correct cannot go down. This is a very concerning methodological misunderstanding and raises serious suspicion as to the validity of the other experiments, too.

**Methods And Evaluation Criteria:**

The dataset, model choice and metrics all make sense to me. In particular, it's good that the authors included leakage-resistant benchmarks like LiveCodeBench. Their findings would have been strengthened somewhat by trying a different teacher model, too, although it is understandable that this may not have been in scope in terms of compute costs.

**Other Comments Or Suggestions:**

None.

**Other Strengths And Weaknesses:**

I appreciate that this paper is, at its core, contributing a non-obvious and somewhat counterintuitive finding: that including the reasoning steps when distilling the ability to generate code from a larger model to a smaller model does not appear to make a measurable difference. That is, to me, a valuable contribution; however, it is hard to tell how much that depends on the teacher model (which they keep fixed).

**Questions For Authors:**

1. Do you think the core finding--that including reasoning is not necessary when distilling from one code llm to another--would translate to different base/teacher models, too?
2. Why do you think the effects of varying the number of samples or the sampling temperature are indicative of the different SFT'd models' underlying capacity?
3. Do you think your supposed finding that putting the reasoning after the code is better than before (or not including it at all) is statistically significant?

**Relation To Broader Scientific Literature:**

The finding that including reasoning steps when distilling from a bigger model to a smaller code-specific model does not seem to benefit downstream performance is somewhat counter-intuitive, and I think an interesting contribution. This goes against the grain of recent findings that suggest that using SFT to distill CoT reasoning discovered with RL into a smaller model can help in math and reasoning domains (c.f. the DeepSeek-R1 paper).

**Theoretical Claims:**

N/A; no theoretical claims or proofs.

---

> ### Author Rebuttal · Authors · 2025-04-01
>
> Thanks for helpful comments! We would like to summarize your concerns and provide our responses below:
>
> 1. **which is the teacher model**: We select **DeepSeek-V2.5-1210** as our teacher model because of its strong capabilities and acceptable cost. It is the most suitable version available during our primary experiment phase.
>
> 2. **different teacher model**: We take **GPT-4o-2024-08-06** as another teacher model to synthesize the CoT&Code dataset again. We perform experiments on DeepSeek-Coder-6.7B-Base using the methods mentioned in the paper and evaluate the model on EvalPlus with the following results:
>
>    |               | HumanEval Base | HumanEval Plus | MBPP base | MBPP Plus | Avg.  |
>    | ------------- | :------------- | -------------- | --------- | --------- | ----- |
>    | $C_{w/o}$     | 71.95          | 65.85          | 77.77     | 67.46     | 70.76 |
>    | $C_{follow}$  | 65.24          | 59.14          | 76.45     | 62.96     | 65.95 |
>    | $C_{precede}$ | 72.56          | 66.46          | 78.57     | 67.72     | 71.33 |
>
> 3. **whether the conclusion contradicts the results of DeepSeek-R1**: Following the release of DeepSeek-R1, we have reviewed our experimental results and believe that our conclusions are indeed consistent with, rather than contradictory, their work. We agree that a proper CoT process is essential for deriving accurate answers, especially in mathematical and logical domains. Instead, we are against the suggestion that CoT processes must be expressed through natural language. Well-designed and high-quality code should also be recognized as a valid form of CoT reasoning, and models do not need to learn to reason through natural language but can learn to code directly and then explain. Thus, our study complements rather than contradicts the results of DeepSeek-R1. To further support this, we take the open source code generation dataset synthesized with **DeepSeek-R1** and directly treat the think and output of R1 as CoT and result, respectively. We fine-tune the models and evaluate them with the same setting in **2**:
>
>    |               | HumanEval Base | HumanEval Plus | MBPP Base | MBPP Plus | Avg.  |
>    | ------------- | :------------- | -------------- | --------- | --------- | ----- |
>    | $C_{w/o}$     | 64.63          | 56.70          | 71.16     | 62.43     | 63.73 |
>    | $C_{follow}$  | 56.09          | 51.82          | 64.55     | 55.02     | 56.87 |
>    | $C_{precede}$ | 64.02          | 57.31          | 71.69     | 62.43     | 63.86 |
>
>    We note that, as mentioned in other recent studies of R1, there is a certain probability that the model will fail to stop thinking when using $C_{follow}$ format.
>
> 4. **whether the evidence sufficiently supports the claim**: We thank the reviewer for raising this point. Indeed, this is consistent with our first argument in the paper, that well-organized and high-quality code can effectively serve as a CoT. Thus, it is reasonable that in some cases the model may have learned enough from the code alone. However, this phenomenon is not universally observed in our experiments. In particular, the explanations following the code added value in most scenarios. Therefore, we hope that the reviewer will notice the second finding, namely that placing the CoT after the code generally provides an additional benefit in most experimental settings, which builds on our initial proposal and is mainly observed. We adhere to standard settings using sufficiently large model and dataset sizes, which ensures stable experimental results even without including variance. Furthermore, we have presented extensive experimental results across various configurations in Appendix G. We appreciate your suggestion and plan to report the statistical result in a future version.
>
> 5. **discussion of pass@k and tempature**: Different k values reflect different aspects of model performance. If model A gives [10, 0, 0] and model B gives [4, 3, 3] over three problems with 10 attempts, both have the same pass@1, but B has a higher pass@10. Small k captures the model's confidence (A solves a problem very well), while larger k reflects the coverage of the model's ability (B can solve more problems). Therefore, varying k reveals different evaluations of model behavior and potential. In addition, temperature also affects the model's performance, with low temperatures tending to make the model's output more fixed and high temperatures encouraging more explore space. We want to show that the conclusions are consistent.
>
> 6. **brief description in discussion**: We apologize for any confusion caused by our brief description of the experiments. We confirm that each experiment was carefully designed with reliable conclusions, and we will make the meaning of the discussions more explicit in future versions.
>
> 7. **missing references**: We appreciate the reviewers for pointing out the omissions in related work, and we promise that these relevant references will definitely be included.

---

> > ### Comment · Reviewer_793T · 2025-04-04
> >
> > Thank you for responding to my questions.
> >
> > After reviewing the reply to my review, as well as those to the others, I have decided to keep my score. The text as is does not sufficiently justify such strong conclusions, in my opinion; as emphasized also by ueRf, without more thorough statistical analysis it is difficult to believe that the results are significant.
> >
> > Furthermore, while I appreciate your reply, it does not explain the discussion that is in the submitted text. In particular, your explanation of why measuring pass@k, which basically boils down to measuring diversity, makes sense. However, the paper just contains this discussion:
> > ```
> > As k increases, all configurations demonstrate an improvement in accuracy, indicating models exploring more possibilities and increasing the chances of producing correct outputs. Among the configurations, C_precede consistently outperforms others, emphasizing the importance of positioning CoT explanation.
> > ```
> > The first observation is then, according to my interpretation of your reply, just saying that the plots suggest that the model has not completely collapsed to a single mode, so drawing more samples leads to increasing (rather than just non-decreasing) pass rates. This I do not believe to be a particularly insightful comment. The second observation is not even correct: looking at the figure, C_w/o has a higher pass@1 than C_precede, for example. This makes your claims appear a bit cherry picked, to be frank.
> >
> > On the whole, I think the experiments are thorough and the findings (if true) are interesting. However, as it stands I do not think the paper does a good enough job providing evidence in their favor. For future versions of this work I would suggest the authors to consider distilling their experiments into a smaller set of findings which are backed up by proper statistical due diligence.

---

> > > ### Author Response · Authors · 2025-04-07
> > >
> > > Thanks for the helpful response! We will summarize your concerns and provide our responses below:
> > >
> > > 1. **Concerns about the discussion of pass@k**: We sincerely thank you for your insightful comments. There was a typo in the presentation that led to loose language:
> > >
> > >    > As k increases, ..., emphasizing the importance of positioning CoT explanations.
> > >
> > >    Our intended message is to claim that different k values correspond to different evaluation settings, allowing us to validate our conclusions' generalizability. To prevent confusion and unnecessary misunderstanding, we will revise it in the next version as:
> > >
> > >    > We change the k-values when evaluating the models to validate the generalizability of our conclusions. The results show that C_precede mostly outperforms the others, highlighting the importance of positioning the CoT explanation.
> > >
> > > 2. **distilling experiments**: All discussions and experiments are designed to support our main conclusion. Because this conclusion is counterintuitive, we conduct experiments from multiple perspectives to elucidate its underlying rationale. Due to space constraints, our explanations are somewhat brief. We will reorganize the discussions and provide additional clarifications to avoid misunderstandings in the future.
> > >
> > > 3. **Significance Analysis:** We fully agree about the importance of significance analysis, and conduct a series of experiments to validate our conclusions. We evaluate the performance of the model on the EvalPlus benchmark and use t-test to show the statistical significance.
> > >
> > >    - We run five experiments on the synthetic dataset with different random seeds and fine-tune the DeepSeek-Coder-6.7B-Base model:
> > >
> > >      | Seed          | 0     | 1     | 2     | 3     | 4     | Avg.       |
> > >      | ------------- | ----- | ----- | ----- | ----- | ----- | ---------- |
> > >      | $C_{w/o}$     | 69.19 | 70.06 | 70.34 | 70.48 | 70.83 | 70.17±0.62 |
> > >      | $C_{follow}$  | 64.88 | 66.54 | 66.03 | 66.38 | 65.84 | 65.93±0.65 |
> > >      | $C_{precede}$ | 71.49 | 71.18 | 71.82 | 71.14 | 70.96 | 71.31±0.34 |
> > >
> > >      The p-values (0.0069, 0.0000) are < 0.05, showing that $C_{precede}$ has a significant performance improvement over the baselines $C_{w/o}$ and $C_{follow}$.
> > >
> > >    - We run five experiments on the synthetic dataset with different random seeds and fine-tune the Llama-3.1-8B-Base model:
> > >
> > >      | Seed          | 0     | 1     | 2     | 3     | 4     | Avg.       |
> > >      | ------------- | ----- | ----- | ----- | ----- | ----- | ---------- |
> > >      | $C_{w/o}$     | 60.47 | 60.98 | 60.6  | 60.61 | 59.84 | 60.50±0.41 |
> > >      | $C_{follow}$  | 58.79 | 59.38 | 60.1  | 59.1  | 59.65 | 59.40±0.51 |
> > >      | $C_{precede}$ | 61.68 | 60.96 | 60.67 | 62.01 | 61.23 | 61.30±0.54 |
> > >
> > >      The p-values (0.0290, 0.0004) are < 0.05, showing that $C_{precede}$ has a significant performance improvement over the baselines $C_{w/o}$ and $C_{follow}$.
> > >
> > >    - We take **GPT-4o-2024-08-06** as another teacher model to synthesize the dataset again. We run five experiments on the new dataset with different random seeds and fine-tune the DeepSeek-Coder-6.7B-Base model:
> > >
> > >      | Seed          | 0     | 1     | 2     | 3     | 4     | Avg.       |
> > >      | ------------- | ----- | ----- | ----- | ----- | ----- | ---------- |
> > >      | $C_{w/o}$     | 70.76 | 70.89 | 69.96 | 71.51 | 71.2  | 70.86±0.58 |
> > >      | $C_{follow}$  | 65.95 | 66.98 | 66.67 | 65.93 | 66.39 | 66.38±0.46 |
> > >      | $C_{precede}$ | 71.33 | 72.4  | 72.18 | 71.96 | 72.58 | 72.09±0.48 |
> > >
> > >      The p-values  (0.0068, 0.0000) are < 0.05, showing that $C_{precede}$ has a significant performance improvement over the baselines $C_{w/o}$ and $C_{follow}$.
> > >
> > >    - We directly use educational instructions from OpenCoder to synthesize the dataset without giving the reference code. We run five experiments on the new dataset with different random seeds and fine-tune the DeepSeek-Coder-6.7B-Base model:
> > >
> > >      | Seed          | 0     | 1     | 2     | 3     | 4     | Avg.       |
> > >      | ------------- | ----- | ----- | ----- | ----- | ----- | ---------- |
> > >      | $C_{w/o}$     | 70.5  | 70.45 | 70.04 | 70.36 | 70.54 | 70.38±0.20 |
> > >      | $C_{follow}$  | 66.29 | 66.94 | 66.13 | 67.44 | 65.54 | 66.47±0.74 |
> > >      | $C_{precede}$ | 71.17 | 72.34 | 70.93 | 71.23 | 71.84 | 71.50±0.58 |
> > >
> > >      The p-values (0.0033, 0.0000) are < 0.05, showing that $C_{precede}$ has a significant performance improvement over the baselines $C_{w/o}$ and $C_{follow}$.

---

### Official Review · Reviewer_ueRf · 2025-03-10

**Overall Recommendation:** 3

**Summary:**

This paper researches on the order of CoT and code in code generation tasks. The authors collect pairs of program questions and response codes in six code data, and then prompt an LM to generate CoT and finally create a dataset with question, code and natural language CoT triplets. Their experiments show that the (question, code, CoT) (that is code precede) surpasses (question, CoT, code) (that is code follow), and they also have many observation about the difference between these two patterns. shed light into how to train a code generation model.

## Update after rebuttal

Experiments with larger models, longer training stage, other CoT source and additional study about the difficulty have partly solved my concerns but I am still not sure whether the conclusions are still correct for most powerful models (like models with >100B parameters and billions of training tokens).

**Claims And Evidence:**

The main claim of the paper is that the code-precede CoT outperforms code-follow CoT in code generation tasks. This claim is supported by experiments on various datasets (e.g., HumanEval, MBPP, LiveCodeBench) and models (e.g., DeepSeek-Coder-6.7B, Llama-3.1-8B, Qwen2.5-chat-7B, Qwen2.5-coder-7B). While the results are convincing, with a consistent and large gap between the two approaches, I believe the absence of variance analysis in the figures and tables is a minor issue. Including such analysis would make the results more robust.

However, I find the authors' explanation "code itself is a suitable CoT, while traditional CoT should be regarded as an explanation of the code rather than reasoning" to be insufficiently supported. I have several concerns:

1. **Task Difficulty**: While the pretrained code corpus is diverse and the LeetCode experiments discuss about different difficulty levels, it's unclear whether the difficulty of the code generation tasks is sufficiently varied. Task complexity ranges from simple translation of instructions to code, to solving complex algorithmic problems or ambiguous GitHub issues. I believe task difficulty significantly impacts the role of code and CoT. For simpler tasks, CoT may act mainly as a post-explanation, but for more complex tasks, defining the problem clearly, choosing the right algorithm, and debugging are crucial parts of the CoT, and these cannot be fully replaced by the code itself. Although the authors provide experiments on Leetcode with varying difficulty, they focus only on code problem complexity. Additional factors like the clarity or completeness of the question, or code length, should also be considered. At the same time, all the four method shows <10% performance on the Leetcode Hard subset, this results may not be very convincing.

2. **Nature of the Phenomenon**: The authors argue that "code itself is a suitable CoT", but it is unclear whether this phenomenon is inherent to code generation tasks or a result of differing difficulty between generating code and CoT. Since code is a formal language, it might be easier for the model to learn than natural language CoT. Further supporting my concerns, Figure 3 shows that the perplexity (ppl) on CoT tokens is significantly higher than that on code tokens, both in the code-precede and code-follow models. This suggests the model is less familiar with natural language CoT. In such cases, CoT might not be helpful for the model. If so, it is doubtful whether a sufficiently powerful model would have yielded different conclusions. After all, being harder does not necessarily mean ultimately worse performance.

3. **Training Dataset and Process**: The training dataset is relatively small (51,200 samples), and the training process only involves SFT. This is much smaller than real-world datasets. For example, DeepSeek-Coder-Instruct uses 2 billion tokens to fine-tune natural language abilities. The paper does not address whether the conclusions would still hold with a larger training set, or if the code-follow approach might perform better with more training data.

4. **Bias from Synthetic CoT**: The synthetic CoT in the experiments is generated by LLM, conditioned on the question and code, which makes it more of an explanation of the code rather than a reasoning process. The authors do not explore whether the results would change if the CoT were generated based on the question and before the code generation, or if it were generated as a human-like reasoning process. In real-world scenarios, CoT typically consists of two parts: one before the code, offering general instructions on how to solve the problem, and one after, explaining the code in detail. The former is more like a reasoning process, and the latter is a code explanation. I believe this distinction should serve as a baseline for comparison.

**Essential References Not Discussed:**

The related work should be updated with some more recent works. For example, though the authors use qwen2.5-code in their experiments, they do not mention the qwen2.5-code in the related work. Some recent works like repo-level code generation (like [SWE-bench](https://www.swebench.com/)) or oi-competition (like [deepseek-r1](https://deepseek.ai/)) should be mentioned. There are some work talking about the CoT and the code like [PAL](https://arxiv.org/abs/2211.10435) and [code-prompt](https://arxiv.org/abs/2305.18507) using code as CoT to solve the math problem. These work are very related to the paper and should be mentioned in the related work.

**Experimental Designs Or Analyses:**

Except for the main results and the concerns I mentioned above. There are many experiments used to explain the different roles of CoT and code in section 6 Discussion. I feel some of them are not necessary, or the authors should explain their motivation more clearly. For example, why the "layer gradient norm", "adversarial code robustness", and "signature removal analysis" are necessary to explain the conclusion. The subsection "Generalization across sizes" can be just moved to the figure 2. The discussion about "temperature" and "pass@k" is too detailed for me and can be moved to the appendix.

And some other experiments need to be explained more clear:

1. In figure 4, the reader cannot understand whether the KL-divergence 0.13 is smaller enough or not.
2. There is a lacking of results analysis of Figure 11.

**Methods And Evaluation Criteria:**

They create datasets with a classic method by using an LLM to generate CoT. There is a concern that the synthetic CoT could introduce bias to the conclusion, as I claimed in the above point 5.

They do not provide a new evaluation metrics but use the commonly used pass@k.

**Other Comments Or Suggestions:**

See aboves.

**Other Strengths And Weaknesses:**

I believe one of the strengths of the article is its extensive experimental content. However, this also raises some concerns for me, as it may be a sign of insufficient core contributions in the paper. This is especially true when considering that many of the experiments in the main text seem somewhat redundant or overly detailed. I think the author should reorganize the three-page discussion more carefully, clearly explaining the motivation, results, and relationships of each experiment. Particularly, if certain experiments are not tightly linked to the conclusions, they should either be removed or placed in the appendix. Based on the current conclusions of the paper, I feel that the contribution falls slightly short of what would be expected from a complete conference paper. The overly detailed experiments, in my opinion, do not add significant value to the overall contribution.

**Questions For Authors:**

There are two main concerns. First, to what extent can the conclusions of the paper hold when the training scale is sufficiently large and the model's capabilities are strong enough? This is crucial as it will determine how the conclusions of the paper impact the training and application of related models. Second, regarding the substantial amount of discussion included in the paper, I would like the author to justify its necessity, explaining how these discussions contribute to the core contributions of the paper, rather than merely summarizing the experimental results.

**Relation To Broader Scientific Literature:**

The observation that code-precede CoT surpasses code-follow CoT is useful to train better code generators, which is an important task in the field of AI. However, I worry about whether the conclusion can be generalized to pretraining stage or on a much larger dataset. If the conclusion is due to the current models are more familiar with code than natural language CoT, or the natural language CoT is naturally harder to learn than code, the current conclusion may not hold when the model is finally trained well on CoT.
At the same time, how the conclusion can be used in practice is not clear. Does the conclusion means that the model should be trained to first generate code and then generate CoT? (Or even without CoT? because I find that even the w/o model show not bad performance.) If so, how can we explain the phenomenon that o1-like long reasoning model always show better performance than general ones like gpt-4o?

**Theoretical Claims:**

There is no theoretical claims.

---

> ### Author Rebuttal · Authors · 2025-04-01
>
> We address the concerns as below:
>
> Q1: How well do the paper's conclusions hold with large-scale training and a powerful model?
>
> R1:**Table A. Extent to a larger scale training dataset:**
>
>  We construct a large dataset of 800K samples. We train Llama-3.1-8B and compare it with DeepSeek-R1-Distill-Llama-8B on EvalPlus:
>
> |                        | HumanEval Base | HumanEval Plus | MBPP base | MBPP Plus | Avg.          |
> | ---------------------- | -------------- | -------------- | --------- | --------- | ------------- |
> | DS-R1-Distill-Llama-8B | 76.82          | 70.73          | 71.42     | 59.52     | 69.62         |
> | Code follow CoT        | 74.39          | **70.12**      | 67.98     | 55.55     | 67.01         |
> | Code precede CoT       | **76.82**      | 69.51          | **74.86** | **62.69** | **70.****97** |
>
>  **Table B. Extent to pre-training stage:**
>
>  We take "the-stack" as the source seed to synthesize the dataset of pretraining. We perform experiments on DeepSeek-Coder-6.7B-Base:
>
> |                  | HumanEval Base | HumanEval Plus | MBPP base | MBPP Plus | Avg.      |
> | ---------------- | -------------- | -------------- | --------- | --------- | --------- |
> | Code follow CoT  | 61.58          | 55.48          | 75.13     | 60.31     | 63.12     |
> | Code precede CoT | **69.51**      | **64.63**      | **77.24** | **63.22** | **68.65** |
>
>  **Table C. Extent to more powerful models**
>
>  We conduct experiments with a larger LLM, Qwen2.5-Coder-Base-32B. The results are reported as follows
>
> |                  | HumanEval Base | HumanEval Plus | MBPP base | MBPP Plus | Avg.  |
> | ---------------- | -------------- | -------------- | --------- | --------- | ----- |
> | Code follow CoT  | 85.98          | 79.27          | 87.83     | 71.69     | 81.19 |
> | Code precede CoT | 86.59          | 79.27          | 89.15     | 76.19     | 82.80 |
>
>  The results in Tables A, B, and C show that our conclusion, "CoT following the code can work better," is held when scaling up the training dataset and model size.
>
> Q2: How can the conclusion be applied in practice, and extensive discussions?
>
> R2: A key contribution is that using the "code precedes CoT" pattern outperforms the "code follows CoT" approach: (1) it establishes a more effective paradigm for integrating CoT with code during training. (2) During inference, it can facilitate early stopping by generating only the code, thereby reducing computational cost and latency. All discussions and experiments are designed to support our main conclusion. We will reorganize them.
>
> Q3: Additional questions raised in the comments:
>
>  Q3.1: Performance across various task difficulties (such as various code length):
>
>  R3.1: We took your suggestion and reclassified the questions' difficulty according to the length of the output answers:
>
> |                  | Length < 500 | 500 < Length < 1000 | 1000 < Length < 1500 | Length > 1500 |
> | ---------------- | ------------ | ------------------- | -------------------- | ------------- |
> | Code follow CoT  | 0.0          | 20.17               | 10.64                | 0.0           |
> | Code precede CoT | 25.0         | 29.20               | 19.23                | 9.09          |
>
> And DeepSeek-Coder-Instruc-6.7B got only 9.1 on Leetcode Hard with the same base.
>
>  Q3.2: Nature of the phenomenon & Q3.3: Training data and process:
>
>  R3.2 & R3.3: These questions have been addressed in R1, which shows that our conclusion still holds with a more powerful model and a larger training set.
>
>  Q3.4: Bias from synthetic CoT.
>
>  R3.4: We provide additional experiments where the CoT is generated first. We directly use educational instructions from OpenCoder to synthesize the CoT without the code and hold the same conclusion.
>
> |                  | HumanEval Base | HumanEval Plus | MBPP base | MBPP Plus | Avg.  |
> | ---------------- | -------------- | -------------- | --------- | --------- | ----- |
> | Code follow CoT  | 62.19          | 54.87          | 71.95     | 57.93     | 61.74 |
> | Code precede CoT | 68.29          | 60.36          | 78.57     | 62.16     | 67.34 |
>
> Q3.5: Variance Analysis.
>
> R3.5: We adhere to standard settings using sufficiently large model and dataset sizes, which ensures stable experimental results even without including variance. Furthermore, we have presented extensive experimental results across various configurations in Appendix G. We appreciate your suggestion and plan to report the variance in a future version.
>
> Q3.6: Explain the o1-like model is better than general ones and the relation with your conclusion.
>
> R3.6: A proper CoT process is crucial for deriving accurate answers, particularly in mathematical and logic domains. However, limited research link CoT and Code for code generation. We argue that using code preceding CoT is an effective data paradigm. Well-designed, high-quality code should be recognized as a valid form of CoT. We checked open-source datasets generated by R1 and confirmed the existence of this phenomenon.

---

> > ### Comment · Reviewer_ueRf · 2025-04-04
> >
> > Thank you for the detailed explanation. I still have a few questions and comments:
> >
> > **Code Length and Task Difficulty**:
> >
> > The length of the code is only one aspect of task difficulty. I believe the nature of the task itself is more important. For instance, a competitive programming problem (OI) is generally more challenging than some lengthy data processing codes, even though the former tends to be much shorter in length. The authors should consider that code generation is a very broad field and ensure that their conclusions are not restricted to a single dataset. Generally speaking, code is a modality but not a task, just like natural language. Just as in natural language, translating a sentence is completely different from solving a math problem. “Translating some instruction to code” or answering some common problems is completely different from creative code writing that requires experimentation and thinking.
> >
> > Additionally, even when using code length as a metric for task difficulty, the length of the "ground truth" or "reference" code would be a better measure than the length of the generated code, as there are potential confounders. For example, longer outputs could result from the model’s verbosity rather than the inherent difficulty of the task. I believe the counterintuitive phenomenon where shorter code performs worse than longer code may be due to such confounders.
> >
> > **Dataset Size and Upper Bound**:
> > While 800k samples is a large dataset, I am uncertain whether it is large enough to effectively measure the upper bound of these two methods. As I mentioned earlier, coder models are generally trained on billions of tokens, and this is also true for CoT models. I think the results in Table A are insufficient to conclusively demonstrate that code-precede CoT outperforms code-precede. It’s still possible that code-precede CoT is simply easier for the model to learn and may perform better on smaller datasets. As evidence, we see that the performance difference between code-precede and code-precede CoT has been reduced from 6% to 4% on average, and the code-follow CoT method outperforms code-precede CoT as the training dataset grows. The same trend is evident in Table C, where the gap significantly shrinks compared to the 7B models. This raises questions about the conclusions for stronger models.
> >
> > ** Significance Analysis**:
> > I want to emphasize that reproducing analysis is a critical component of any research, particularly when considering the shrinking gap between the model and the dataset, as mentioned earlier. Ignoring statistic analysis can seriously undermine the credibility of research in the field, as it may lead to misleading results. I encourage the authors to follow established best practices in this regard, such as the [AAAI checklist](https://aaai.org/conference/aaai/aaai-25/aaai-25-reproducibility-checklist/), which includes items like:
> >
> > >This paper states the number of algorithm runs used to compute each reported result. (yes/no)
> > >
> > >Analysis of experiments goes beyond single-dimensional summaries of performance (e.g., average; median) to include measures of variation, confidence, or other distributional information. (yes/no)
> > >
> > >The significance of any improvement or decrease in performance is judged using appropriate statistical tests (e.g., Wilcoxon signed-rank). (yes/partial/no)
> >
> > Similarly, the [ACL checklist](https://aclrollingreview.org/responsibleNLPresearch/) advises:
> >
> > >Did you report descriptive statistics about your results (e.g., error bars around results, summary statistics from sets of experiments), and is it transparent whether you are reporting the max, mean, etc. or just a single run?
> >
> > Without repeated experiments and statistical analysis, I am unsure whether the observed gaps are statistically significant.

---

> > > ### Author Response · Authors · 2025-04-07
> > >
> > > Thanks for the helpful response! We will summarize your concerns and provide our responses below:
> > >
> > > 1. **Dataset Size and Upper Bound**: We use an 800k training dataset in the additional experiment (see Table A), following the configuration of DeepSeek-R1, which also uses 800k SFT data for DeepSeek-R1-Distill-Llama-8B. Thus, this setting is both reasonable and large enough to effectively validate our conclusions. Furthermore, the significance analysis results show a statistically significant improvement from our method, confirming that the amount of training data is adequate. In addition, there is no solid evidence that the improvement converges with the scale of the training dataset and the size of the model. Although the results may show some trends due to the capability of the model, the significance analysis has shown that the benefit of our method remains robust as the dataset and model scale.
> > >
> > > 2. **Code Length and Task Difficulty**: We take your suggestion and classify the questions' difficulty according to the length of the reference code:
> > >
> > >    |                  | Length < 500 | 500 < Length < 1000 | 1000 < Length < 1500 | Length > 1500 |
> > >    | ---------------- | ------------ | ------------------- | -------------------- | ------------- |
> > >    | Code follow CoT  | 21.42        | 13.26               | 11.11                | 0.0           |
> > >    | Code precede CoT | 42.85        | 17.34               | 27.77                | 33.33         |
> > >
> > >    As for competitive programming problem, we choose LiveCodeBench with difficulies, collecting problems from contests across competition platforms, like CodeForces:
> > >
> > >    |                  | Easy  | Medium | Hard |
> > >    | ---------------- | ----- | ------ | ---- |
> > >    | Code follow CoT  | 45.19 | 9.51   | 0.74 |
> > >    | Code precede CoT | 49.67 | 9.72   | 1.04 |
> > >
> > > 3. **Significance Analysis:** We fully agree about the importance of significance analysis, and conduct a series of experiments to validate our conclusions. We evaluate the performance of the model on the EvalPlus benchmark and use t-test to show the statistical significance.
> > >
> > >    - We run five experiments on the synthetic dataset with different random seeds and fine-tune the DeepSeek-Coder-6.7B-Base model:
> > >
> > >      | Seed          | 0     | 1     | 2     | 3     | 4     | Avg.       |
> > >      | ------------- | ----- | ----- | ----- | ----- | ----- | ---------- |
> > >      | $C_{w/o}$     | 69.19 | 70.06 | 70.34 | 70.48 | 70.83 | 70.17±0.62 |
> > >      | $C_{follow}$  | 64.88 | 66.54 | 66.03 | 66.38 | 65.84 | 65.93±0.65 |
> > >      | $C_{precede}$ | 71.49 | 71.18 | 71.82 | 71.14 | 70.96 | 71.31±0.34 |
> > >
> > >      The p-values (0.0069, 0.0000) are < 0.05, showing that $C_{precede}$ has a significant performance improvement over the baselines $C_{w/o}$ and $C_{follow}$.
> > >
> > >    - We run five experiments on the synthetic dataset with different random seeds and fine-tune the Llama-3.1-8B-Base model:
> > >
> > >      | Seed          | 0     | 1     | 2     | 3     | 4     | Avg.       |
> > >      | ------------- | ----- | ----- | ----- | ----- | ----- | ---------- |
> > >      | $C_{w/o}$     | 60.47 | 60.98 | 60.6  | 60.61 | 59.84 | 60.50±0.41 |
> > >      | $C_{follow}$  | 58.79 | 59.38 | 60.1  | 59.1  | 59.65 | 59.40±0.51 |
> > >      | $C_{precede}$ | 61.68 | 60.96 | 60.67 | 62.01 | 61.23 | 61.30±0.54 |
> > >
> > >      The p-values (0.0290, 0.0004) are < 0.05, showing that $C_{precede}$ has a significant performance improvement over the baselines $C_{w/o}$ and $C_{follow}$.
> > >
> > >    - We take **GPT-4o-2024-08-06** as another teacher model to synthesize the dataset again. We run five experiments on the new dataset with different random seeds and fine-tune the DeepSeek-Coder-6.7B-Base model:
> > >
> > >      | Seed          | 0     | 1     | 2     | 3     | 4     | Avg.       |
> > >      | ------------- | ----- | ----- | ----- | ----- | ----- | ---------- |
> > >      | $C_{w/o}$     | 70.76 | 70.89 | 69.96 | 71.51 | 71.2  | 70.86±0.58 |
> > >      | $C_{follow}$  | 65.95 | 66.98 | 66.67 | 65.93 | 66.39 | 66.38±0.46 |
> > >      | $C_{precede}$ | 71.33 | 72.4  | 72.18 | 71.96 | 72.58 | 72.09±0.48 |
> > >
> > >      The p-values  (0.0068, 0.0000) are < 0.05, showing that $C_{precede}$ has a significant performance improvement over the baselines $C_{w/o}$ and $C_{follow}$.
> > >
> > >    - We directly use educational instructions from OpenCoder to synthesize the dataset without giving the reference code. We run five experiments on the new dataset with different random seeds and fine-tune the DeepSeek-Coder-6.7B-Base model:
> > >
> > >      | Seed          | 0     | 1     | 2     | 3     | 4     | Avg.       |
> > >      | ------------- | ----- | ----- | ----- | ----- | ----- | ---------- |
> > >      | $C_{w/o}$     | 70.5  | 70.45 | 70.04 | 70.36 | 70.54 | 70.38±0.20 |
> > >      | $C_{follow}$  | 66.29 | 66.94 | 66.13 | 67.44 | 65.54 | 66.47±0.74 |
> > >      | $C_{precede}$ | 71.17 | 72.34 | 70.93 | 71.23 | 71.84 | 71.50±0.58 |
> > >
> > >      The p-values (0.0033, 0.0000) are < 0.05, showing that $C_{precede}$ has a significant performance improvement over the baselines $C_{w/o}$ and $C_{follow}$.

---

### Official Review · Reviewer_PRDj · 2025-03-14

**Overall Recommendation:** 4

**Summary:**

This paper primarily investigates how chain-of-thought reasoning affects code generation performance. The paper first constructs a dataset of 50k pairs for code generation. A series of experiments is then conducted to investigate how the presence and position of a chain-of-thought affect resulting code generation performance.

On HumanEval, BigCodeBench, and LiveCodeBench; the authors find that the standard procedure of having code follow the CoT is in many cases substantially worse than having the CoT follow the code. A series of experiments with different model families (DeepSeek, Llama, Qwen-Chat/Coder) show similar results.

There are a large number of additional experiments ablating and studying this effect in various settings.

**Claims And Evidence:**

Yes.

**Essential References Not Discussed:**

None.

**Experimental Designs Or Analyses:**

I checked both Section 5 and Section 6.

**Methods And Evaluation Criteria:**

Yes.

**Other Comments Or Suggestions:**

None.

**Other Strengths And Weaknesses:**

I think the main finding of the paper is very interesting and throws into question the useful of chain-of-thought for code generation. In addition, the experiments are well done, and there is a comprehensive discussion section that includes a large number of ancillary results.

**Questions For Authors:**

None.

**Relation To Broader Scientific Literature:**

I do not think the specific research question addressed by the authors has been explored by prior work. I think the closest papers to this paper are papers like Program-of-Thoughts, which demonstrate that reasoning in code can be superior to reasoning in natural language. A common assumption in most reasoning papers is that CoT should come before the final answer, but I don't think any paper has shown the counterintuitive empirical result that the CoT is better off coming after the code.

**Theoretical Claims:**

Not applicable.

---

> ### Author Rebuttal · Authors · 2025-04-01
>
> Thanks for helpful comments! We appreciate the reviewers for pointing out some other reference in related work, and we promise that these relevant references will definitely be included in the future revision.

---

### Official Review · Reviewer_gzpe · 2025-03-15

**Overall Recommendation:** 3

**Summary:**

The paper argues that in the context of fine tuning a code generation LLM, appending the CoT after the code solution works better compared to the typical setting of prepending it. In order to show this they generate 50k code generation problems with code solutions and CoTs. Experimentally, they demonstrate SFT on training data with CoT preceding code degrades performance, while CoTs following code results in improvements.

**Claims And Evidence:**

The primary claim in the paper is that the best order for a CoT coding SFT dataset is: prompt -> code -> CoT. The paper provides ample evidence that this applies to the specific dataset the authors created as part of this submission. This is a caveat that should be better highlighted and discussed in the paper. In particular, the question remains whether the findings generalize to SFT data with CoTs that are generated through a different template or generally with a different process.

**Essential References Not Discussed:**

not that I'm aware of

**Experimental Designs Or Analyses:**

Some ablations in the discussion section only seem tangentially related and could benefit from some further explanation:
- context attention weight plots: what's the x axis? why's the split between code and CoT the same region between left and right? How come both split between NLP and Code/CoT are equally large?
- layer gradient norm: what's the significance of this result? Why does it matter that the gradients of the two datasets have different magnitudes? how does that compare to other datasets where the order of the CoT isn't relevant or works better in the standard order?
- inconsistent data analysis, adversarial code robustness: how are these relevant to the main claim of the paper?

**Methods And Evaluation Criteria:**

Appending CoTs to the code instead of prepending it is sensible when the CoT is viewed as an explanation. The dataset used for the main results (LiveCodeBench) is the current gold standard.

**Other Comments Or Suggestions:**

the Figure 1 legend is hard to read. maybe make the lines thicker and add the symbols to the legend?

**Other Strengths And Weaknesses:**

strengths:
- the paper points out an interesting result in the context of SFT for code generation
- the experimental results and analysis are quite extensive

weaknesses:
- more care and discussion should have been devoted to the relation between the specific training dataset and the experimental findings. In particular, appending CoT to code appears to only slightly improve over code without CoT (if at all), while CoT before code appears to significantly degrade the performance. the latter result is surprising and can benefit from more analysis and discussion

**Questions For Authors:**

My primary concern is that it's unclear whether these findings will generalize to other training datasets. As mentioned in the previous comments, I'd like to understand:
1. will other training data distributions yield the same results or is there a specific characteristic for the CoTs that make them more useful when appended?
2. why does appending the CoT degrade the performance so significantly?

**Relation To Broader Scientific Literature:**

Currently, inducing CoTs before the solution is the predominant method to make use of extra tokens and also the way in which CoT SFT data is constructed. This paper challenges this approach in the context of code generation.

**Theoretical Claims:**

no theory

---

> ### Author Rebuttal · Authors · 2025-04-01
>
> Thanks for the helpful comments! We will summarize your concerns and provide our responses below. We explored many other training data distributions:
>
> 1. **different teacher model**: We take **GPT-4o-2024-08-06** as another teacher model to synthesize the CoT&Code dataset again. We perform experiments on DeepSeek-Coder-6.7B-Base using the methods mentioned in the paper and evaluate the model on EvalPlus with the following results:
>
>    |               | HumanEval Base | HumanEval Plus | MBPP base | MBPP Plus | Avg.  |
>    | ------------- | :------------- | -------------- | --------- | --------- | ----- |
>    | $C_{w/o}$     | 71.95          | 65.85          | 77.77     | 67.46     | 70.76 |
>    | $C_{follow}$  | 65.24          | 59.14          | 76.45     | 62.96     | 65.95 |
>    | $C_{precede}$ | 72.56          | 66.46          | 78.57     | 67.72     | 71.33 |
>
> 2. **different training stage**: We take **the-stack** dataset from pre-training stage to synthesize the dataset for the pre-training stage. We fine-tune the models and evaluate them with the same setting in **(1)**:
>
>    |               | HumanEval Base | HumanEval Plus | MBPP base | MBPP Plus | Avg.  |
>    | ------------- | :------------- | -------------- | --------- | --------- | ----- |
>    | $C_{w/o}$     | 68.29          | 61.58          | 76.98     | 62.43     | 67.32 |
>    | $C_{follow}$  | 61.58          | 55.48          | 75.13     | 60.31     | 63.13 |
>    | $C_{precede}$ | 69.51          | 64.63          | 77.24     | 63.22     | 68.65 |
>
> 3. **different synthetic template**: We direct use educational instructions from OpenCoder to synthesize the dataset without giving the reference code. We fine-tune the models and evaluate them with the same setting in **(1)**:
>
>    |               | HumanEval Base | HumanEval Plus | MBPP base | MBPP Plus | Avg.  |
>    | ------------- | :------------- | -------------- | --------- | --------- | ----- |
>    | $C_{w/o}$     | 67.07          | 60.97          | 76.71     | 60.31     | 66.27 |
>    | $C_{follow}$  | 62.19          | 54.87          | 71.95     | 57.93     | 61.74 |
>    | $C_{precede}$ | 68.29          | 60.36          | 78.57     | 62.16     | 67.35 |
>
> 4. **different dataset source**: We take the code generation dataset synthesized with Deepseek-R1 from the open source website, and directly treat the think and output of R1 as CoT and result, respectively. We fine-tune the models and evaluate them with the same setting in **(1)**:
>
>    |               | HumanEval Base | HumanEval Plus | MBPP Base | MBPP Plus | Avg.  |
>    | ------------- | :------------- | -------------- | --------- | --------- | ----- |
>    | $C_{w/o}$     | 64.63          | 56.70          | 71.16     | 62.43     | 63.73 |
>    | $C_{follow}$  | 56.09          | 51.82          | 64.55     | 55.02     | 56.87 |
>    | $C_{precede}$ | 64.02          | 57.31          | 71.69     | 62.43     | 63.86 |
>
> 5. **CoT precedes code degrading performance**: In this case, the model has to learn the teacher's reasoning without prior knowledge. This step tends to be more difficult for the base model because natural language distribution is more diverse. In particular, the base model often struggles to learn directly the thought generated by another strong model.
>
> 6. **discussion of attention weight**: The length of each sequence (instruction, code, and CoT) is different, we first extract the attention matrix of the last layer and then partition it into 3×3 sub-blocks using a bilinear 2D interpolation method. We normalize the attention matrix across the validation set to ensure that the attention weights are comparable across rows. Therefore, the x-axis of the resulting plot represents the sequential input of tokens from the concatenated input, with its division into three equal segments reflecting the equal-sized attention weight matrices between the instruction, code, and chain-of-thought parts.
>
> 7. **discussion of layer gradient norm**: The gradient norms of different layers in the right plot are generally smaller and show a smoother, more gradual transition between layers. Previous research suggests that smooth gradient norm distributions during training can reduce the risk of unstable optimization, resulting in more stable and consistent parameter updates.
>
> 8. **discussion of inconsistent data**: This discussion aims to investigate the conditions under which high-quality code data can serve as a CoT. Our results in the inconsistent data experiment suggest that self-consistency alone is not a reliable indicator of code quality since the teacher model may generate the wrong test cases, and the generated code may not be incorrect.
>
> 9. **discussion of adversarial code**: The experiments investigate whether lexical information affects the quality of the code. Our results show that model performance suffers significantly under such perturbations, suggesting that current models often rely heavily on lexical representations for semantic understanding.

---

### Decision · Program_Chairs · 2025-05-01

**Decision:**

Accept (poster)

**Comment:**

This work looks into question of tuning a smaller model for NL code generation using CoT traces. Surprisingly, they find that tuning a model to first produce the code and then the CoT yields improved models compared to the “traditional” option of first using the CoT and then the code. An extensive evaluation (some in the rebuttal text) evaluates this effect and ablates various parameters.

Overall, this seems like an interesting and unexpected result that the ICML community should be aware of and hence I recommend that this work is accepted. I would kindly ask the authors to update their work to include the extensive results shown in rebuttal.